# 3-in-1: 2D Rotary Adaptation for Efficient Finetuning, Efficient Batching and Composability

**Baohao Liao**[1,2*]   **Christof Monz**[1]
[1]Language Technology Lab, University of Amsterdam
[2]eBay Inc., Aachen, Germany
Code: https://github.com/BaohaoLiao/road

## Abstract

Parameter-efficient finetuning (PEFT) methods effectively adapt large language models (LLMs) to diverse downstream tasks, reducing storage and GPU memory demands. Despite these advantages, several applications pose new challenges to PEFT beyond mere parameter efficiency. One notable challenge involves the efficient deployment of LLMs equipped with multiple task- or user-specific adapters, particularly when different adapters are needed for distinct requests within the same batch. Another challenge is the interpretability of LLMs, which is crucial for understanding how LLMs function. Previous studies introduced various approaches to address different challenges. In this paper, we introduce a novel method, RoAd, which employs a straightforward 2D rotation to adapt LLMs and addresses all the above challenges: (1) RoAd is remarkably parameter-efficient, delivering optimal performance on GLUE, eight commonsense reasoning tasks and four arithmetic reasoning tasks with $< 0.1\%$ trainable parameters; (2) RoAd facilitates the efficient serving of requests requiring different adapters within a batch, with an overhead comparable to element-wise multiplication instead of batch matrix multiplication; (3) RoAd enhances LLM's interpretability through integration within a framework of distributed interchange intervention, demonstrated via composition experiments.

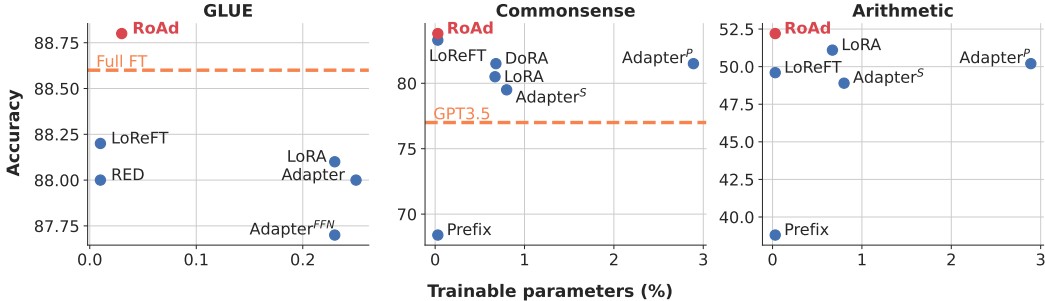

Figure 1: Performance of various PEFT methods on the GLUE benchmark, eight commonsense reasoning tasks and four arithmetic reasoning tasks with RoBERTa-large or LLaMA-13B.

## 1 Introduction

Large language models (LLMs), trained on extensive web-scale datasets to perform tasks such as predicting masked words [8, 31, 45] or anticipating the next word in a sentence [17, 52, 53],

---

*Correspondence to b.liao@uva.nl. Please go to https://arxiv.org/abs/2409.00119 for the newest version.

38th Conference on Neural Information Processing Systems (NeurIPS 2024).

demonstrate remarkable effectiveness across a range of NLP applications. For tasks where the data distribution diverges from that of the pretraining corpus, finetuning emerges as an effective way to tailor an LLM to specific requirements. Leveraging the capabilities of LLMs, recent studies [13, 14, 22, 23, 25, 27, 42, 60, 62, 65] demonstrate that training only a subset of an LLM's parameters can yield performance on par with full finetuning. This approach, termed parameter-efficient finetuning (PEFT), provides two primary advantages: (1) It reduces the storage requirements for trained parameters, as it necessitates preserving only a universal LLM alongside a minimal set of task-specific parameters; (2) It decreases GPU memory consumption during finetuning, owing to the reduction in optimizer state sizes which correlate directly with the number of trainable parameters.

With the evolution of PEFT, concerns extend beyond mere parameter efficiency. PEFT encounters a variety of challenges brought forth by diverse applications. A significant challenge is the efficient deployment of personalized or task-specific LLMs [25, 57]. These applications frequently require distinct sets of trained parameters for different tasks or users. When multiple users submit requests simultaneously, it becomes crucial to process these requests collectively in a single batch. Given that each request may require a unique set of parameters, using batch matrix multiplication can efficiently handle these requests by leveraging GPU parallelism. However, the batch matrix multiplication still incurs considerable overhead [1, 57], necessitating the exploration of more efficient methods.

Another challenge is the interpretability of LLMs that contain a billion-scale of parameters, making it difficult to explore their mechanism. PEFT provides an alternative approach by constraining the number of trainable parameters, thereby aiding in interpretability. Recent advancements in PEFT methods, particularly those focusing on representation editing [54, 60, 67], can be incorporated within an intervention framework [11]. This integration enhances their capability for interpretability, offering a more manageable means of dissecting the operational intricacies of LLMs.

In this paper, we introduce a novel technique termed 2D rotary adaptation (RoAd) which efficiently adapts LLMs using a minimal number of trainable parameters. Furthermore, RoAd enhances both batching efficiency and composability. Our initial investigation reveals that finetuning primarily alters the angular components of the representations in pretrained LLMs, rather than their magnitudes (Section §3.1). Based on this observation, we employ a strategy of rotating certain subspaces within the representations to emulate finetuning effects. Specifically, we implement a 2D rotational approach on the representations and develop three distinct variants of RoAd (Section §3.2).

To assess the efficacy of RoAd, we perform comprehensive evaluations on the GLUE benchmark [56], eight commonsense reasoning tasks and four arithmetic reasoning tasks, utilizing RoBERTa [31] and LLaMA [52, 53] (Section §4.1). The results consistently show that RoAd surpasses other PEFT methods while maintaining a significantly reduced scale of trainable parameters ($< 0.1\%$), as depicted in Figure 1. Additionally, RoAd employs element-wise rather than matrix multiplication, which notably improves throughput when serving heterogeneous requests within the same batch, achieving twice the throughput of LoRA [14] (Section §4.2). Furthermore, RoAd can be seamlessly integrated within an intervention framework [11], thereby enhancing model interpretability. We illustrate this through a composition experiment, demonstrating RoAd's capacity to merge weights trained for different tasks and display a new capability (Section §4.3).

## 2   Background

In this section, we outline the challenges tackled in this work, illustrating the constraints of existing methods and objectives that drive the development of the proposed method, RoAd.

### 2.1   Parameter-efficient finetuning (PEFT)

Existing PEFT techniques can be categorized into three groups: adapter-based, prompt-based, and latency-less methods. Adapter-based methods [12, 13, 42] incorporate adapters either in parallel with or sequentially to the existing Transformer [55] modules. This incorporation necessitates modifications to the LLM architecture, consequently adding extra latency during inference. Prompt-based methods [19, 21, 43] enhance the input by appending new trainable tokens, which lengthens the sequence and thereby increases the computational overhead during inference. Latency-less methods, such as LoRA [14] and its variants [22, 27, 65], apply low-rank matrices to adapt the pretrained weights. These matrices can be seamlessly integrated into the existing weight matrices following

finetuning, thus preserving the original LLM architecture. Specifically, LoRA adapts an LLM as $\boldsymbol{W} = \boldsymbol{W}^0 + \Delta\boldsymbol{W}$, where $\boldsymbol{W}^0 \in \mathbb{R}^{d_1 \times d_2}$ is the pretrained weight and $\Delta\boldsymbol{W} = \boldsymbol{B}\boldsymbol{A}$ with $\boldsymbol{B} \in \mathbb{R}^{d_1 \times r}$, $\boldsymbol{A} \in \mathbb{R}^{r \times d_2}$, $r \ll d_1$ and $r \ll d_2$. Our proposed method, RoAd, aligns with the latency-less category and integrates effortlessly into the existing linear layer without imposing additional overhead during inference. Moreover, RoAd demonstrates exceptional parameter efficiency. The quantity of its trainable parameters is equivalent to that of a LoRA module with a rank $r = 0.5$.

**Orthogonal finetuning.** Drawing on the concept of hyperspherical energy and its role in characterizing generalization [28, 29], OFT [44] introduces orthogonal finetuning, an effective PEFT method for finetuning text-to-image diffusion models. Specifically, OFT implements an orthogonal matrix $\boldsymbol{R} \in \mathbb{R}^{d_1 \times d_1}$ to the pretrained weight $\boldsymbol{W}^0$, so the input $\boldsymbol{x} \in \mathbb{R}^{d_1}$ to a linear layer after adaptation becomes $\boldsymbol{z} = (\boldsymbol{R}\boldsymbol{W}^0)^\top \boldsymbol{x}$. $\boldsymbol{R}$ is parameter-efficient because it is a block-diagonal matrix with $n$ blocks as $\boldsymbol{R} = \mathrm{diag}(\boldsymbol{R}_1, ..., \boldsymbol{R}_i, ..., \boldsymbol{R}_n)$, where each block $\boldsymbol{R}_i \in \mathbb{R}^{w \times w}$ has a dimension $w = d_1/n$. To maintain orthogonality, $\boldsymbol{R}_i$ is derived using Cayley parameterization: $\boldsymbol{R}_i = (\boldsymbol{I} + \boldsymbol{Q}_i)(\boldsymbol{I} - \boldsymbol{Q}_i)^{-1}$ with $\boldsymbol{Q}_i \in \mathbb{R}^{w \times w}$ being a skew-symmetric matrix ($\boldsymbol{Q}_i = -\boldsymbol{Q}_i^\top$). In sum, $\{\boldsymbol{Q}_i\}_{i=1}^n$ serve as the trainable parameters and $\boldsymbol{R}$ is constructed from them with Cayley parameterization. Subsequent advancement, BOFT [30], leverages butterfly factorization to further refine OFT's parameter efficiency. However, both OFT and BOFT, due to their reliance on matrix inversions in the Cayley parameterization and increased storage of intermediate activations, necessitate additional GPU memory and increase training duration compared to other PEFT approaches. Conversely, RoAd, which may be considered as a specialized case of OFT with $w = 2$, offers a faster and more memory-efficient solution by inherently maintaining orthogonality without requiring further parameterization.

## 2.2 Batching

Batching in this context refers to processing multiple heterogeneous requests, each requiring different adapters[2] for inference. This scenario commonly arises when serving personalized or task-specific LLMs. Specifically, we consider a setup where distinct adapters instead of a shared adapter are finetuned for various tasks to achieve optimal performance. During inference, each request in a batch pertains to a different task and necessitates a unique adapter.

Consider that we have finetuned distinct LoRA modules for $b$ tasks, denoted as $\{\boldsymbol{A}_i, \boldsymbol{B}_i\}_{i=1}^b$. For a batch of $b$ requests represented as $\boldsymbol{X} \in \mathbb{R}^{b \times l \times d_1}$, where $l$ is the maximum sequence length across the requests, each request requires a different LoRA module. To exploit the parallel processing capabilities of GPUs, the output $\boldsymbol{Z}$ of a linear layer can be computed as follows: First, the output from the pretrained layer is computed as $\boldsymbol{Z}^0 = torch.mm(\boldsymbol{X}, \boldsymbol{W}^0)$. Subsequently, the intermediate output from the first low-rank matrix, $\hat{\boldsymbol{B}} \in \mathbb{R}^{b \times d_1 \times r}$ (a concatenation of $\{\boldsymbol{B}_i\}_{i=1}^b$), is obtained as $\boldsymbol{Z}_0^1 = torch.bmm(\boldsymbol{X}, \hat{\boldsymbol{B}})$. The output from the second low-rank matrix, $\hat{\boldsymbol{A}} \in \mathbb{R}^{b \times r \times d_2}$ (a concatenation of $\{\boldsymbol{A}_i\}_{i=1}^b$), follows as $\boldsymbol{Z}^1 = torch.bmm(\boldsymbol{Z}_0^1, \hat{\boldsymbol{A}})$. Finally, these outputs are summed to produce $\boldsymbol{Z} = \boldsymbol{Z}^0 + \boldsymbol{Z}^1$. It is noteworthy that batch matrix multiplication (BMM), as implemented in $torch.bmm$, often introduces substantial overhead [1], reducing throughput and increasing latency, which adversely impacts user experience in time-sensitive applications.

In contrast, prompt-based methods circumvent the use of BMM by appending trainable tokens to each request, simplifying the computational process. However, prompt-based methods with long prompt tokens are difficult to optimize, which degrades performance compared to other PEFTs [14, 15]. $(\text{IA})^3$ [25] proposes adapting LLM by multiplying the output from a linear layer with a trainable vector, involving only element-wise multiplication for efficient batching. A recent development, FLoRA [58], builds on $(\text{IA})^3$ by employing two low-rank matrices while maintaining element-wise operations. Although our proposed method, RoAd, requires BMM, its sparse structure allows a reformulation of BMM and results in an overhead equivalent to element-wise multiplication.

## 2.3 Intervention and composability

Numerous studies [10, 11, 37, 38, 40] have provided support for the linear representation hypothesis [35, 46, 49] that concepts are represented within linear subspaces of neural network representations. To examine if a concept is captured within a linear subspace of a representation, Geiger et al. [11]

---

[2]Adapter here means the trained parameters since LoRA's architecture is also similar to an adapter.

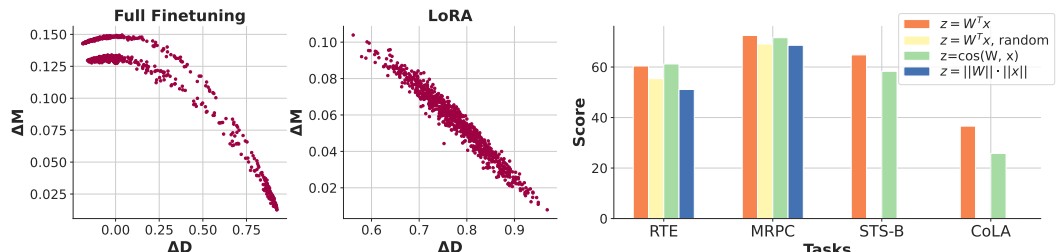

Figure 2: Pilot study for the pretrained and finetuned representations. **Left & Middle**: The change in magnitude and angle of representations between pretrained and finetuned LLM using full finetuning or LoRA. **Right**: The disentanglement experiment of magnitude and angle of pretrained representation.

suggests employing a distributed interchange intervention (DII) defined as:

$$\text{DII}(\boldsymbol{b}, \boldsymbol{s}, \boldsymbol{R}) = \boldsymbol{b} + \boldsymbol{R}^\top (\boldsymbol{R}\boldsymbol{s} - \boldsymbol{R}\boldsymbol{b}) \tag{1}$$

$\boldsymbol{b}$ denotes the hidden representation generated at row $i$ and column $k$ when the model processes an input, while $\boldsymbol{s}$ represents the corresponding representation when the model processes a different input. The matrix $\boldsymbol{R} \in \mathbb{R}^{r \times d_1}$, consisting of orthogonal rows, serves as a low-rank projection matrix where $d_1$ is the dimension of the representation and $r$ is the subspace dimension under intervention. Equation (1) illustrates the application of a DII to $\boldsymbol{b}$ using a counterfactual source representation $\boldsymbol{s}$.[3]

Drawing inspiration from this established framework, a recent study, LoReFT [61], introduces a method for finetuning specific positions of the representations to adapt LLM. This study further demonstrates that several prior approaches of representation editing [54, 60, 67] can be effectively integrated within this framework. Interestingly, the application of RoAd to representations can also be conceptualized as DII, offering interpretability potential. To demonstrate one aspect of interpretability for RoAd, we primarily conduct a qualitative experiment focused on task composition. This experiment involves combining the weights of models trained on distinct tasks to showcase the capability for multitasking learning without the need for additional adaptation [16, 20, 61, 64, 66].

## 3   Method

In this section, we first perform two pilot studies to ascertain the key factor influencing the adaptation of LLMs. Following this, we present our proposed method, the 2D rotary adaptation (RoAd), which serves as an effective PEFT method addressing the various challenges outlined in Section §2.

### 3.1   Pilot study

**Study 1: Variations in magnitude and angular displacement.** Assume $\boldsymbol{x}^0, \boldsymbol{x} \in \mathbb{R}^{d_1}$ are representations of the same token from a pretrained and finetuned LLM, respectively. We define the relative change in magnitude as $\Delta M = \left| \|\boldsymbol{x}\|_2 - \|\boldsymbol{x}^0\|_2 \right| / \|\boldsymbol{x}^0\|_2$ and compute the angular displacement as $\Delta D = \cos(\boldsymbol{x}, \boldsymbol{x}^0) \in [-1, 1]$. A larger $\Delta M$ and a smaller $\Delta D$ indicate more significant changes in magnitude and angular displacement, respectively. Our study involves: (1) finetuning RoBERTa-base [31] on the SST-2 task [50] using either full finetuning or LoRA; (2) extracting representations $\boldsymbol{x}^0$ and $\boldsymbol{x}$ from the output of the second-last Transformer block for the [CLS] token across all samples in the development set, followed by computing $\Delta M$ and $\Delta D$.[4] As depicted in Figure 2 (Left and Middle), there is a more pronounced change in $\Delta D$ than in $\Delta M$ for both full finetuning and LoRA.[5]

**Study 2: Disentanglement of magnitude and angle.** To ascertain whether angular or magnitude adjustments are more critical for finetuning, we implement a disentanglement study. This involves freezing RoBERTa-base and appending a two-layer classifier on top of it. The first layer of this

---

[3]We adopt notation systems from Wu et al. [61].

[4]Please refer to Figure B.1 for all layers.

[5]There are two other interesting observations: (1) An increase in magnitude change correlates with a larger angular displacement; (2) Compared to LoRA, full finetuning has a bigger change in magnitude and angle (for all layers, see Figure B.1), which is in line with a recent finding that LoRA learns less and forgets less [2].

classifier incorporates a weight matrix $W \in \mathbb{R}^{d_1 \times d_1}$. Under standard operations, the output from this layer is computed as $z = W^\top x^0$. To distinctly evaluate the impacts of magnitude and angle, we modify the output to retain only the magnitude component as $z_i = \|W_{:,i}\|_2 \cdot \|x^0\|_2$, or solely the angular component as $z_i = \cos(W_{:,i}, x^0)$ ($z_i$ is the i$^{\text{th}}$ element of $z$). The modified classifier was then finetuned on four GLUE tasks with different metrics detailed in Table C.1. Additionally, a weak baseline employing a randomly initialized RoBERTa-base is included. As shown in Figure 2 (Right), angular information is paramount in finetuning, whereas reliance solely on magnitude information even leads to inferior results compared to the random backbone.

Both studies indicate that angular information is more crucial than magnitude information for adapting a pretrained LLM to a downstream task. However, rotating the entire $d_1$ dimensions of the representation for finetuning incurs substantial computational costs. These costs are primarily reflected in a large number of trainable parameters, necessitating a dense matrix $R \in \mathbb{R}^{d_1 \times d_1}$, and in the requirement to maintain its orthogonality. Could we only rotate a subspace of the representation and design a $R$ that is always orthogonal without any parameterization as OFT [44]? The first idea that comes to our mind is 2D rotation which only rotates two dimensions at a time and inherently maintains orthogonality.

### 3.2 2D rotary adaptation

Suppose that $W^0 \in \mathbb{R}^{d_1 \times d_2}$ is the pretrained weight of a linear layer, $x \in \mathbb{R}^{d_1}$ is the input of a token to this linear layer, $R \in \mathbb{R}^{d_2 \times d_2}$ is the rotation matrix, the adapted output from the linear layer is $z = Rh = R(W^{0\top}x)$. The rotation matrix $R$ is defined as follows:

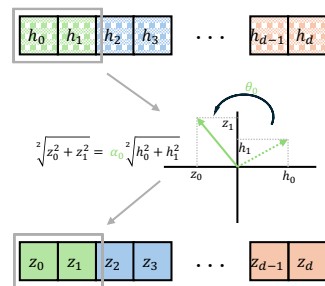

$$R = \text{diag}(R_1, R_2, ..., R_{d_2/2}) \quad \text{with} \quad R_i = \begin{bmatrix} \cos\theta_i & -\sin\theta_i \\ \sin\theta_i & \cos\theta_i \end{bmatrix}$$
(2)

The trainable parameters are denoted as $\{\theta_i\}_{i=1}^{d_2/2}$. This 2D rotary adaptation involves rotating pairs of adjacent dimensions of $h$, specifically dimensions $2i-1$ and $2i$, using the rotation matrix $R_i$.[6] The rotation matrix $R$ is characterized by its parameter efficiency, which

Figure 3: Overview of RoAd$_1$.

is attributed to its sparse structure and the parameter sharing within each block $R_i$. Additionally, $R$ can be integrated directly into the existing pretrained weights, forming $W = W^0 R^\top$, which does not incur additional computational costs during inference. This design closely mirrors RoPE [51], with the notable difference that in our RoAd, $\theta_i$ is trainable and $R_i$ does not incorporate positional information. The overview of RoAd is shown in Figure 3.

**Relaxation to orthogonality.** Referring to Figure 2 (Right), while reliance predominantly on angular information substantially outperforms reliance on magnitude information, it remains less effective than using both angular and magnitude information for the tasks of MRPC, STS-B, and CoLA. Furthermore, both fully- and LoRA-finetuned LLMs exhibit slight adaptations in magnitude, as depicted in Figure 2 (Left and Middle). Consequently, we modify $R_i$ by incorporating $\alpha_i$ to regulate the magnitude. We define a general $R_i$ as follows:

$$R_i = \begin{bmatrix} \alpha_{i,11} \cos\theta_{i,11} & -\alpha_{i,12} \sin\theta_{i,12} \\ \alpha_{i,21} \sin\theta_{i,21} & \alpha_{i,22} \cos\theta_{i,22} \end{bmatrix}$$
(3)

We develop three variants of RoAd by altering the configuration of shared parameters as outlined in Table 1. RoAd$_1$ introduces a minimal change to Equation (2) by incorporating a scaling factor $\alpha_i$. RoAd$_1$ already shows impressive results for most tasks in Section §4.1. For some knowledge-intensive tasks, we observe that RoAd$_2$ and RoAd$_4$ obtain better results with more trainable parameters. To preserve the starting point of LLMs [23], we always initialize $\alpha_i = 1$ and $\theta_i = 0$.

**Batching.** In practice, we don't need to save $R$ as a sparse matrix and do matrix multiplication. Taking RoAd$_1$ as an example in Equation (4), we only save two vectors: $R^1$ and $R^2$. Then $z = Rh = R^1 \otimes h + R^2 \otimes \hat{h}$, where $\hat{h}$ is a rearranged version of $h$ and $\otimes$ denotes element-wise multiplication. This reformulation not only simplifies the representation of $R$ but also enhances the efficiency of batching in RoAd, relying solely on element-wise multiplications rather than BMM.

---

[6]The index in this work starts from 1 instead of 0.

Table 1: A summarization of three RoAd variants.

| RoAd$_?$ | $\alpha_i$ | $\theta_i$ | #Trainable |
|---|---|---|---|
| 1 | $\alpha_{i,11} = \alpha_{i,12} = \alpha_{i,21} = \alpha_{i,22} = \alpha_i$ | $\theta_{i,11} = \theta_{i,12} = \theta_{i,21} = \theta_{i,22} = \theta_i$ | $d_2$ |
| 2 | $\alpha_{i,11} = \alpha_{i,12} \quad \alpha_{i,21} = \alpha_{i,22}$ | $\theta_{i,11} = \theta_{i,12} \quad \theta_{i,21} = \theta_{i,22}$ | $2d_2$ |
| 4 | $\alpha_{i,11} \neq \alpha_{i,12} \neq \alpha_{i,21} \neq \alpha_{i,22}$ | $\theta_{i,11} \neq \theta_{i,12} \neq \theta_{i,21} \neq \theta_{i,22}$ | $4d_2$ |

$$
\boldsymbol{z} = \boldsymbol{R}\boldsymbol{h} = \boldsymbol{R}^1 \otimes \boldsymbol{h} + \boldsymbol{R}^2 \otimes \hat{\boldsymbol{h}}
$$

$$
= \begin{bmatrix} \alpha_1 \cos\theta_1 \\ \alpha_1 \cos\theta_1 \\ \alpha_2 \cos\theta_2 \\ \alpha_2 \cos\theta_2 \\ \vdots \\ \alpha_{d_2/2} \cos\theta_{d_2/2} \\ \alpha_{d_2/2} \cos\theta_{d_2/2} \end{bmatrix} \otimes \begin{bmatrix} h_1 \\ h_2 \\ h_3 \\ h_4 \\ \vdots \\ h_{d_2-1} \\ h_{d_2} \end{bmatrix} + \begin{bmatrix} \alpha_1 \sin\theta_1 \\ \alpha_1 \sin\theta_1 \\ \alpha_2 \sin\theta_2 \\ \alpha_2 \sin\theta_2 \\ \vdots \\ \alpha_{d_2/2} \sin\theta_{d_2/2} \\ \alpha_{d_2/2} \sin\theta_{d_2/2} \end{bmatrix} \otimes \begin{bmatrix} -h_2 \\ h_1 \\ -h_4 \\ h_3 \\ \vdots \\ -h_{d_2} \\ h_{d_2-1} \end{bmatrix} \tag{4}
$$

**Composability.** RoAd can be incorporated into the DII framework as $\Phi(\boldsymbol{h}) = \boldsymbol{R}\boldsymbol{h} = \boldsymbol{h} + \boldsymbol{R}(\boldsymbol{h} - \boldsymbol{R}^\top \boldsymbol{h})$, with $\boldsymbol{R}\boldsymbol{s}$ in Equation (1) being set to $\boldsymbol{h}$. Although a degree of relaxation is introduced to the orthogonality of $\boldsymbol{R}$, it is important to note that the rows of $\boldsymbol{R}$ remain orthogonal to each other within non-adjacent segments of the same block, $\boldsymbol{R}_i$. This offers a possibility for composability. We can finetune some rows on one task and other orthogonal rows on another task. Since they are orthogonal to each other, these two tasks should minimally affect each other, and the combination of these rows after finetuning could bring new multitasking learning ability.

RoAd can be considered as a special case of OFT [44] with $w = 2$. However, it is much more parameter- and memory-efficient and faster. Please refer to Section §D.1 for a detailed discussion.

## 4 Experiments

In this section, we begin by implementing RoAd to finetune various LLMs across three benchmarks. Subsequently, we illustrate its efficiency in batching processes and demonstrate its composability. Unless otherwise noted, RoAd is applied to all linear layers within the LLMs. All of our experiments are conducted on A100 80GB GPU with the frameworks, Transformers [59] and PEFT [34].

### 4.1 Results on downstream tasks

**Natural language understanding (NLU).** We evaluate the effectiveness of RoAd on the GLUE benchmark [56] for its ability of NLU with RoBERTa [31] as the backbone. Unlike many previous works [14, 22, 23, 31, 65] that employ the GLUE development sets for both validation and testing, here we partition the development set into distinct validation and test subsets to mitigate the risk of overfitting. For comprehensive information regarding the split of the development set, the search space of hyperparameters, the optimal hyperparameter configurations, and other details crucial for reproducibility, please see Section §C.1.

As shown in Table 2, RoAd$_1$ outperforms all other PEFT methods with $< 0.1\%$ trainable parameters for both sizes of RoBERTa on average, being the only PEFT method that matches or outperforms full finetuning. These results show that 2D rotation (with a few scaling) can efficiently adapt LLM.

**Commonsense reasoning.** In assessing the capacity of LLaMA [52] for commonsense reasoning, we focus on eight representative tasks: BoolQ [4], PIQA [3], SIQA [48], HellaSwag [63], WinoGrande [47], ARC-e, ARC-c [5], and OBQA [36]. The setting here contrasts with the NLU experiments where each task involves finetuning a separate LLM. Instead, we adopt a unified strategy by finetuning a single LLM across all tasks as delineated in Hu et al. [15]. Such a setting is designed to mitigate overfitting and aligns more closely with real-world applications. Specifically, the training and test sets from these eight tasks are reformulated according to a predefined template, so all tasks can be trained or evaluated in a generative way. For all finetuning experiments on LLaMA, we follow a recipe in Table C.5 without extensive searching. Please see Section §C.2 for more training details.

Table 2: Results on the held-out GLUE development set with RoBERTa as the backbone. We report matched accuracy for MNLI, Matthew's correlation for CoLA, Pearson correlation for STS-B and accuracy for other tasks. The **best** and second-best results are in bold and underlined, respectively, being the same for other tables. The percentage of trainable parameters is calculated without considering the classifier head. $RoAd_1(fc1)$ means that we only insert the $RoAd_1$ module to the first feed-forward layer, to match the #Params. of RED and LoReFT. Results of methods denoted by $*$ and $\diamond$ are from Wu et al. [60] and Wu et al. [61], respectively. Otherwise, average results from three random runs are reported. Refer to Table C.4 for the standard deviation.

| Model | Method | #Params. | RTE | MRPC | STS-B | CoLA | SST-2 | QNLI | QQP | MNLI | Avg. |
|---|---|---|---|---|---|---|---|---|---|---|---|
| | Full FT* | 100.00% | 78.3 | 87.9 | 90.6 | 62.4 | 94.4 | 92.5 | 91.7 | 87.3 | 85.6 |
| base | Adapter* | 0.32% | 76.5 | 88.4 | **90.5** | 60.9 | 93.3 | 92.5 | **90.5** | 87.0 | 85.0 |
| | LoRA* | 0.24% | 75.3 | 88.7 | 90.3 | 59.7 | 93.9 | 92.6 | 90.4 | 86.6 | 84.7 |
| | Adapter$^{FNN}$* | 0.24% | 77.7 | 88.8 | 90.4 | 58.5 | 93.0 | 92.0 | 90.2 | **87.1** | 84.7 |
| | BOFT | 0.16% | 71.4 | 87.5 | 89.6 | 55.3 | 92.5 | 91.4 | 89.4 | 85.3 | 82.8 |
| | OFT$_{w=2}$ | 0.10% | 74.4 | 87.6 | 89.4 | 50.4 | 92.8 | 90.9 | 89.2 | 83.9 | 82.3 |
| | BitFit* | 0.08% | 69.8 | 88.0 | 89.5 | 54.0 | 94.0 | 91.0 | 87.3 | 84.7 | 82.3 |
| | (IA)$^3$ | 0.04% | 75.3 | 87.1 | 90.0 | 60.4 | 94.0 | 91.8 | 89.2 | 85.8 | 84.2 |
| | RED* | 0.02% | 78.0 | 89.2 | 90.4 | 61.0 | 93.9 | 90.7 | 87.2 | 83.9 | 84.3 |
| | LoReFT$^\diamond$ | 0.02% | 79.0 | 89.2 | 90.0 | 60.4 | 93.4 | 91.2 | 87.4 | 83.1 | 84.2 |
| | **RoAd$_1$** | 0.07% | 78.9 | 89.2 | 90.5 | 64.4 | 93.9 | 91.9 | 89.6 | 86.3 | **85.6** |
| | **RoAd$_1$**(fc1) | 0.03% | **79.1** | **90.2** | 90.2 | 60.9 | 94.6 | 91.6 | 88.7 | 85.4 | 85.1 |
| | Full FT* | 100.00% | 85.8 | 91.7 | 92.6 | 68.2 | 96.0 | 93.8 | 91.5 | 88.8 | 88.6 |
| large | Adapter* | 0.25% | 85.3 | 90.5 | 91.5 | 65.4 | 95.2 | 94.6 | **91.4** | 90.1 | 88.0 |
| | LoRA* | 0.23% | 86.3 | 89.8 | 91.7 | 65.5 | 96.0 | **94.7** | 90.7 | 90.1 | 88.1 |
| | Adapter$^{FNN}$* | 0.23% | 84.8 | 90.5 | 90.2 | 64.4 | 96.1 | 94.3 | 91.3 | **90.3** | 87.7 |
| | RED* | 0.01% | 86.2 | 90.3 | 91.3 | **68.1** | 96.0 | 93.5 | 88.8 | 89.5 | 88.0 |
| | LoReFT$^\diamond$ | 0.01% | 87.5 | 90.1 | 91.6 | 68.0 | 96.2 | 94.1 | 88.5 | 89.2 | 88.2 |
| | **RoAd$_1$** | 0.06% | **89.2** | 91.0 | 91.7 | 66.1 | **96.3** | 94.4 | 91.0 | 89.7 | 88.7 |
| | **RoAd$_1$**(fc1) | 0.03% | 88.7 | **91.5** | 91.9 | 68.1 | 96.1 | 94.5 | 90.2 | 89.6 | **88.8** |

As shown in Table 3, RoAds still perform the best across various PEFT methods for both LLaMA-7B and LLaMA-13B on average. The strong baseline to RoAd is a recent representation finetuning method, LoReFT [61], 80.2 vs. 79.2 and 83.3 vs. 83.0 for RoAd$_1$ for LLaMA-7B and LLaMA-13B, respectively. With a slightly increasing number of trainable parameters from RoAd$_1$ to RoAd$_2$ or RoAd$_4$, RoAd matches or outperforms LoReFT. The same story is also told for another two versions of LLaMA, i.e. LLaMA2 [53] and LLaMA3, in Table D.2.

**Arithmetic reasoning.** To assess the arithmetic reasoning ability of LLMs, we evaluate the finetuned LLMs on the test sets of four tasks: AQuA [24], GSM8K [6], MAWPS [18] and SVAMP [41]. Similar to the commonsense reasoning tasks, we finetune a single LLM for all four arithmetic reasoning tasks. The training dataset is Math10K [15] which is constructed from the training sets of GSM8K, MAWPS, MAWPS-single and AQuA. The training recipe is similar to the one used for commonsense reasoning as shown in Table C.5. Please see Section §C.3 for more training details.

Different from the results of NLU and commonsense reasoning tasks, RoAd doesn't always perform the best on the arithmetic reasoning tasks, as shown in Figure 4. For the smaller-size LLM, LLaMA-7B, RoAd is significantly better than other PEFT methods with $< 0.1\%$ trainable parameters, but worse than LoRA and Adapter$^P$ with more than $10\times$ trainable parameters. However, for the larger-size LLM, LLaMA-13B, all RoAd variants are better than other PEFT methods, which shows its scalability and potentially implies even better results for larger LLMs.

Observed from the above-mentioned results, for enhanced performance on downstream tasks and if a marginal increase in the storage capacity for trained parameters is acceptable, RoAd$_4$ is the preferable option. Conversely, if the objective is to investigate how the model adjusts in terms of angle and magnitude, RoAd$_1$ is recommended. Notably, all variants of RoAd incur the same computational overhead for batching.

Table 5: Score on AlpacaEval2.0 with LLaMA2-7B.

| Method | #Params. | Finetuning Data | Win Rate (%) |
|---|---|---|---|
| LoRA | 0.83% | 10K cleaned Alpaca | 61.55 |
| LoReFT | 0.03% | 10K cleaned Alpaca | 60.21 |
| RoAd$_1$ | 0.02% | 10K cleaned Alpaca | **62.64** |
| LoReFT | 0.03% | UltraFeedback [7] | 61.68 |
| RoAd$_1$ | 0.02% | UltraFeedback | **62.60** |

Table 3: Accuracy of LLaMA on eight commonsense reasoning tasks. Results of methods denoted by *, $\diamond$ and $\circ$ are from [15], [61] and [27], respectively. Otherwise, average results from three random runs are reported. Refer to Table C.6 for the standard deviation. Refer to Table D.2 for LLaMA2&3.

| Model | Method | #Paras. | BoolQ | PIQA | SIQA | HellaS. | WinoG. | ARC-e | ARC-c | OBQA | Avg. |
|---|---|---|---|---|---|---|---|---|---|---|---|
| GPT3.5* | - | - | 73.1 | 85.4 | 68.5 | 78.5 | 66.1 | 89.8 | 79.9 | 74.8 | 77.0 |
| 7B | Adapter[P]* | 3.54% | 67.9 | 76.4 | 78.8 | 69.8 | 78.9 | 73.7 | 57.3 | 75.2 | 72.3 |
| | Adapter[S]* | 0.99% | 63.0 | 79.2 | 76.3 | 67.9 | 75.7 | 74.5 | 57.1 | 72.4 | 70.8 |
| | DoRA$^\circ$ | 0.84% | 68.5 | 82.9 | _79.6_ | 84.8 | 80.8 | 81.4 | 65.8 | _81.0_ | 78.1 |
| | LoRA* | 0.83% | 68.9 | 80.7 | 77.4 | 78.1 | 78.8 | 77.8 | 61.3 | 74.8 | 74.7 |
| | OFT | 0.14% | 69.0 | 82.0 | 78.5 | 90.9 | 78.9 | 83.0 | 68.2 | 76.4 | 78.4 |
| | Prefix* | 0.04% | 64.3 | 76.8 | 73.9 | 42.1 | 72.1 | 72.9 | 54.0 | 60.6 | 64.6 |
| | LoReFT$^\diamond$ | 0.03% | 69.3 | **84.4** | 80.3 | **93.1** | **84.2** | 83.2 | 68.2 | 78.9 | **80.2** |
| | (IA)$^3$ | 0.02% | 67.8 | 81.7 | 78.1 | 89.9 | 81.1 | 80.5 | 65.4 | 77.8 | 77.8 |
| | **RoAd$_4$** | 0.08% | **70.6** | _83.2_ | 79.0 | _92.3_ | 81.8 | **84.2** | **70.6** | 80.0 | **80.2** |
| | **RoAd$_2$** | 0.04% | 70.3 | 82.6 | 79.2 | 92.0 | _81.8_ | **84.8** | 68.8 | **82.2** | **80.2** |
| | **RoAd$_1$** | 0.02% | _70.4_ | 81.9 | 79.0 | 91.4 | 80.3 | 84.0 | 68.7 | 77.8 | _79.2_ |
| 13B | Adapter[P]* | 2.89% | 72.5 | 84.9 | 79.8 | 92.1 | 84.7 | 84.2 | 71.2 | 82.4 | 81.5 |
| | Adapter[S]* | 0.80% | 71.8 | 83.0 | 79.2 | 88.1 | 82.4 | 82.5 | 67.3 | 81.8 | 79.5 |
| | DoRA$^\circ$ | 0.68% | 72.4 | 84.9 | 81.5 | 92.4 | 84.2 | 84.2 | 69.6 | 82.8 | 81.5 |
| | LoRA* | 0.67% | 72.1 | 83.5 | 80.5 | 90.5 | 83.7 | 82.8 | 68.3 | 82.4 | 80.5 |
| | Prefix* | 0.03% | 65.3 | 75.4 | 72.1 | 55.2 | 68.6 | 79.5 | 62.9 | 68.0 | 68.4 |
| | LoReFT$^\diamond$ | 0.03% | 72.1 | _86.3_ | 81.8 | **95.1** | 87.2 | 86.2 | 73.7 | 84.2 | 83.3 |
| | **RoAd$_4$** | 0.07% | _73.2_ | 85.5 | **82.4** | _94.5_ | 86.3 | 86.8 | **74.6** | 86.0 | _83.7_ |
| | **RoAd$_2$** | 0.03% | **73.3** | **86.4** | _82.0_ | 94.4 | 86.1 | **87.4** | 74.1 | **87.0** | **83.8** |
| | **RoAd$_1$** | 0.02% | 72.2 | 85.1 | 81.2 | 94.1 | 84.4 | 86.6 | 73.7 | _86.6_ | 83.0 |

Table 4: Accuracy of LLaMA on four arithmetic reasoning tasks. Results of methods denoted by * and $\diamond$ are from [15] and [61], respectively. Refer to Table C.7 for the standard deviation.

| Model | Method | #Params. | AQuA | GSM8K | MAWPS | SVAMP | Avg. |
|---|---|---|---|---|---|---|---|
| 7B | Adapter[P]* | 3.54% | 18.1 | _35.3_ | **82.4** | 49.6 | _46.4_ |
| | Adapter[S]* | 0.99% | 15.0 | 33.3 | 77.7 | _52.3_ | 44.6 |
| | LoRA* | 0.83% | 18.9 | **37.5** | 79.0 | _52.1_ | **46.9** |
| | Prefix* | 0.04% | 14.2 | 24.4 | 63.4 | 38.1 | 35.0 |
| | LoReFT$^\diamond$ | 0.03% | 21.4 | 26.0 | 76.2 | 46.8 | 42.6 |
| | (IA)$^3$ | 0.02% | 19.7 | 28.8 | 76.9 | 48.5 | 43.5 |
| | **RoAd$_4$** | 0.08% | 24.8 | 27.4 | _81.5_ | 49.4 | 45.8 |
| | **RoAd$_2$** | 0.04% | **26.8** | 29.9 | 78.6 | 49.3 | 46.2 |
| | **RoAd$_1$** | 0.02% | **26.4** | 26.2 | 76.5 | 46.7 | 44.0 |
| 13B | Adapter[P]* | 2.89% | 20.5 | 43.3 | 81.1 | 55.7 | 50.2 |
| | Adapter[S]* | 0.80% | 22.0 | _44.0_ | 78.6 | 50.8 | 48.9 |
| | LoRA* | 0.67% | 18.5 | **47.5** | 83.6 | 54.6 | 51.1 |
| | Prefix* | 0.03% | 15.7 | 31.1 | 66.8 | 41.4 | 38.8 |
| | LoReFT$^\diamond$ | 0.03% | 23.6 | 38.1 | 82.4 | 54.2 | 49.6 |
| | **RoAd$_4$** | 0.07% | _25.2_ | 39.8 | _84.5_ | **59.5** | **52.3** |
| | **RoAd$_2$** | 0.03% | **26.0** | 40.6 | 84.0 | _58.3_ | _52.2_ |
| | **RoAd$_1$** | 0.02% | 24.8 | 40.7 | **84.9** | 57.3 | 51.9 |

**Instruction-following ability.** We further benchmark RoAd using AlpacaEval2.0 [9]. We finetune LLaMA2-7B with two instruction-tuning datasets and evaluate the model using AlpacaEval2.0. This evaluation employs GPT-4 [39] to assess the responses generated by the finetuned model against those produced by Text-davinci-003. We don't choose GPT-4 as the reference model, because GPT-4 is too powerful than LLaMA2-7B. The proof-of-concept experiment with LoRA shows the win-rate $< 5\%$. As shown in Table 5, RoAd$_1$ demonstrates superior performance compared to all baselines, while utilizing the least number of trainable parameters.

**Multimodal ability.** Lastly, we apply RoAd to the LLM backbone of LLaVA [26]. Liu et al. [26] requires 4.61% trainable parameters for LoRA on this task, while most tasks with LoRA in our paper need $< 1\%$, showing that this task is knowledge-intensive. Therefore, we need to scale RoAd's trainable parameters. For this purpose, we combine it with LoRA due to the limited number of $\theta_i$ and $\alpha_i$ in $\boldsymbol{R}$. The combination is represented as $\boldsymbol{z} = (\boldsymbol{R}\boldsymbol{W}^{0\top} + (\boldsymbol{B}\boldsymbol{A})^{\top})\boldsymbol{x}$, where $\boldsymbol{A}$ and $\boldsymbol{B}$ are from LoRA. We adjust the LoRA rank to vary the number of trainable parameters. We combine RoAd$_1$

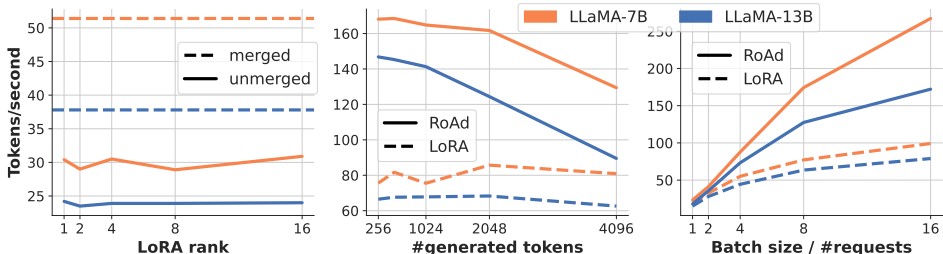

Figure 4: Comparison of throughput between LoRA and RoAd. **Left**: The influence of weight merging for LoRA. **Middle**: The influence of the number of generated tokens. **Right**: The influence of the number of heterogeneous requests in a batch.

with LoRA, but not $RoAd_2$ or $RoAd_4$, as their primary design purpose is to increase the number of trainable parameters.

As shown in Table 6, with only 0.08% trainable parameters, $RoAd_4$ already achieves 96.9% of the accuracy of LoRA with 4.61% trainable parameters. By combining $RoAd_1$ with LoRA, we achieve the same performance as LoRA with only 1/4 of its trainable parameters. This demonstrates RoAd's excellent scalability when combined with LoRA.

Table 6: Visual instruction tuning results on LLaVA1.5-7B.

| Method | #Params. | GQA | SQA | VQAT | POPE | Avg. |
|---|---|---|---|---|---|---|
| LoRA | 4.61% | 62.4 | **68.5** | 56.9 | **86.0** | **68.5** |
| $RoAd_4$ | 0.08% | 60.0 | 66.9 | 53.3 | 85.5 | 66.4 |
| $RoAd_1$ + LoRA | 1.19% | **62.5** | 68.2 | **57.4** | 85.8 | **68.5** |

## 4.2 Efficiency results for batching

We commence by highlighting the significance of weight merging for PEFT. Among the approaches discussed in Section §4.1, only LoRA [14], DoRA [27], BOFT [30], OFT [44], BitFit [62], (IA)³ [25], and our proposed RoAd enable the integration of trainable parameters with pretrained parameters without incurring additional inference overhead. As an illustration, we consider LoRA both with and without weight merging to underscore this process's importance. Notably, the implementation of LoRA with merged weights effectively reverts to the original LLM. To assess throughput, we configure the system with a batch size of 1, generate 2048 tokens, and apply the LoRA modules across all linear layers. Figure 4 (Left) clearly illustrates that the unmerged LoRA exhibits a significantly smaller throughput compared to the merged LoRA. Additionally, it is evident that the throughput of the unmerged LoRA demonstrates only a weak correlation with the rank size, primarily due to the fact that the additional overhead is largely attributed to communication instead of computation.

Furthermore, to evaluate the throughput of batching, we establish a default batch size of 8, generate 2048 tokens, and set the LoRA rank to 8. Each request within the batch is heterogeneous, necessitating eight distinct sets of trainable parameters by default. We only compare to LoRA here, because other baselines have either a weaker performance on downstream tasks (BOFT, OFT, BitFit and (IA)³) or a smaller throughput than LoRA for batching (DoRA). As shown in Figure 4 (Middle and Right), RoAd significantly outperforms LoRA with variations in either the number of generated tokens or the number of heterogeneous requests. With an increasing number of distinct requests, the gap between LoRA and RoAd becomes even larger, which shows RoAd's unique advantage in efficient serving

## 4.3 Qualitative results for composability

In our investigation of RoAd's ability to handle compositional tasks, we primarily engage in multilingual experiments similar to those conducted by Wu et al. [61]. We use two training datasets: a new version of HellaSwag [63][7], which comprises 1K samples with prompts in English and completions in German, and a 1K-sample subset of the Ultrafeedback [7] dataset, which focuses on instruction following tasks in English. Contrary to the above experiments that adapt the outputs of the linear layer, here we instead adapt the representations from the $16^{th}$ block of LLaMA-7B, treating RoAd as a DII method. Specifically, we only adapt/intervene the representation of the final token in the

---
[7]https://huggingface.co/datasets/LeoLM/HellaSwag_de

> **LoReFT response for prompt: Do you know the conference NeurIPS?**
>
> **German subspace**: geht auf dem Bildschirm vorbei und zeigt die Konferenz. (Translation: passes by on the screen and shows the conference.)   ✗
> **Instruction following subspace**: NeurIPS (Neural Information Processing Systems) is an annual conference on machine learning and computational neuroscience ...   ✓
> **Combined subspaces**: Es gibt eine Konferenz namens NeurIPS, die in der Vergangenheit stattgefunden hat. (Translation: There is a conference called NeurIPS that took place in the past.)   ✗

> **RoAd response for prompt: Do you know the conference NeurIPS?**
>
> **German subspace**: Ich weiß nicht, ob ich es kenne. (Translation: I don't know if I know it.)   ✗
> **Instruction following subspace**: Yes, I do! It's a great conference for machine learning researchers and practitioners to share their work with each other.   ✓
> **Combined subspaces**: NeurIPS ist ein internationales Forschungskongress, der sich mit dem Thema künstliche Intelligenz beschäftigt. (Translation: NeurIPS is an international research congress that deals with the topic of artificial intelligence.)   ✓

Figure 5: Qualitative comparison between RoAd and LoReFT for their composability. The prompt is always in English for different subspaces. Refer to Figure D.1, D.2 and D.3 for more examples.

prompt using RoAd$_1$. We train the upper half of $\boldsymbol{R}$, i.e. $\{\boldsymbol{R}_i\}_{i=1}^{d_2/4}$, to handle the German completions in HellaSwag, and another half to complete the English sentences in Ultrafeedback. Both tasks are simultaneously trained but utilize distinct subspaces of $\boldsymbol{R}$. We train the model over five epochs with a learning rate of $5e-3$ and a batch size of $8$.[8]

As in Figure 5, both LoReFT and RoAd are unable to perform completions with the German subspace. This limitation is anticipated due to two primary reasons: (1) LLaMA-7B predominantly relies on pretraining from English datasets, and doesn't have a cross-lingual answering ability without explicitly prompting. (2) The HellaSwag dataset is relatively small, containing only 1K samples with limited comprehensive coverage. Despite these constraints, the German subspace effectively prompts the model to produce sentences in German. Additionally, both methods achieve accurate completions in the other half of the subspaces, attributed to LLaMA-7B's extensive knowledge base in English. When these two subspaces are combined, RoAd successfully leverages their strengths, facilitating accurate sentence completions in German, while LoReFT doesn't catch the purpose of the prompt. We offer more examples, including negative examples, in Figure D.1, D.2 and D.3.

## 5   Conclusion

Initially, our research examines how finetuning modifies the representation of pretrained LLMs, finding that angular adjustments are more significant than changes in magnitude scale. Leveraging this insight, we propose a PEFT method, RoAd, which primarily utilizes a 2D rotational adjustment to the representation. Despite its simplicity, RoAd exhibits several distinct advantages: (1) It is exceptionally efficient in terms of parameters, consistently delivering superior performance on downstream tasks with the fewest trainable parameters compared to other PEFT methods; (2) RoAd efficiently supports batch processing, achieving twice the throughput of LoRA; (3) When incorporated within an intervention framework, RoAd demonstrates remarkable composability.

Due to page limit, we discuss the limitations and broader impacts in Section §A and §B, respectively.

## Acknowledgements

We thank eBay Inc. for the computation support. This research was funded in part by the Netherlands Organization for Scientific Research (NWO) under project number VI.C.192.080.

---

[8]The experiment is based on this notebook `https://github.com/stanfordnlp/pyreft/blob/main/examples/composition/compreft.ipynb`.

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

# A    Limitations

We recognize that a primary limitation pertains to the scalability of RoAd. Currently, it is not feasible to indefinitely increase the number of trainable parameters with RoAd. Nevertheless, our experiments demonstrate that RoAd$_4$ already exhibits commendable performance. To scale the trainable parameters, we can combine RoAd with other PEFT methods, such as LoRA, which enhances the scaling behavior of these PEFTs, i.e. achieving similar results with less trainable parameters.

# B    Broader impacts

RoAd's primary advantage is its efficiency in adapting LLMs to specific tasks with minimal trainable parameters. This efficiency not only reduces computational resource needs but also makes advanced AI technologies more accessible to organizations with limited resources, potentially democratizing AI capabilities across smaller enterprises and educational institutions. By reducing the number of trainable parameters and the computational load, RoAd likely decreases the energy consumption associated with training and deploying LLMs. This could contribute to lowering the carbon footprint of AI research and deployment, aligning with greater environmental sustainability efforts. The ability to process multiple heterogeneous requests efficiently means that applications can provide personalized, context-specific responses more quickly. This enhances the user experience in real-time applications, such as digital assistants, automated service, and interactive educational platforms.

While RoAd improves interpretability in some aspects by integrating within frameworks like distributed interchange intervention [11], the overall complexity of the methods might still pose challenges in understanding and diagnosing the models' decisions. This could affect efforts to make AI more transparent and accountable, especially in critical applications like healthcare and law. Increasing the accessibility of powerful AI models through PEFT also raises concerns about misuse. More entities can harness these capabilities, potentially including those with malicious intents, such as creating sophisticated disinformation campaigns or automating cyber attacks.

# C    Experimental details

## C.1    Natural language understanding (NLU)

Table C.1: The data statistics and evaluation metrics of the GLUE benchmark. The valid and test sets are randomly split from the original development set. Following Wu et al. [60], only the matched development set of MNLI is used. For runs with different seeds, the samples in the valid and test sets are also different.

| Task | RTE | MRPC | STS-B | CoLA | SST-2 | QNLI | QQP | MNLI |
|---|---|---|---|---|---|---|---|---|
| **#Train** | 2.6K | 3.7K | 5.7K | 8.5K | 67K | 105K | 364K | 393K |
| **#Valid** | 139 | 204 | 750 | 522 | 436 | 1K | 1K | 1K |
| **#Test** | 138 | 204 | 750 | 521 | 436 | 4.5K | 39K | 8K |
| **Metric** | Acc. | Acc. | Pearson | Matthew | Acc. | Acc. | Acc. | Acc. |

**Test set split.** Previous works [14, 22, 31] report the best results on the development sets of the GLUE tasks, i.e. using the same set for both validation and test, which might cause overfitting. Instead, we follow the setting of Mahabadi et al. [33] and Wu et al. [60], splitting the whole development set into a validation set and a test set. The model with the best performance on the validation set is selected to perform on the test set. Specifically, for the task with a development set whose number of samples is larger than 2K, i.e. QNLI, QQP and MNLI, we randomly select 1K samples as the validation set and the rest as the test set. For the other tasks, we select half of the samples in the development set as the validation set and another half as the test set. Please refer to Table C.1 for more details.

**Hyperparameter tuning.** We mainly follow the hyperparameter search space of Liao et al. [22] and list them in Table C.2. Notably, we almost upscale the learning rate by 10 for RoAd, because RoAd prefers a larger learning rate than other PEFT methods, which is also observed from Liu et al. [25] and Wen and Chaudhuri [57] where their adapters also apply multiplication instead of addition. The

Table C.2: Hyperparameter search space for GLUE. For tasks with a large number of training samples, we set the number of epochs as 10. Please refer to Table C.3 for the best task-specific settings.

| Hyperparameters | RTE, MRPC, STS-B, CoLA | SST-2, QNLI, QQP, MNLI |
|---|---|---|
| Optimizer | AdamW | AdamW |
| Weight decay | 0 | 0 |
| LR | {1e-3, 3e-3, 5e-3, 7e-3} | {1e-3, 3e-3, 5e-3, 7e-3} |
| LR scheduler | Linear | Linear |
| Warmup ratio | 0.1 | 0.1 |
| Epochs | {10, 20} | 10 |
| Batch size | {16, 32} | {16, 32} |

Table C.3: Best hyperparameter settings for different GLUE tasks on RoBERTa. Notably, RoAd has a very consistent recipe for different tasks. The low-resource tasks (RTE, MRPC, STS-B, CoLA) and high-resource tasks (SST-2, QNLI, QQP, MNLI) show two obvious patterns for the hyperparameters. If you have enough computation resources, we suggest alternating the batch size of low-resource tasks (RTE, MRPC, STS-B, CoLA) in {16, 32} and the number of epochs in {10, 20}, since these tasks have a relatively larger variance.

| Model | Hyperparameter | RTE | MRPC | STS-B | CoLA | SST-2 | QNLI | QQP | MNLI |
|---|---|---|---|---|---|---|---|---|---|
| base | LR | 3e-3 | 3e-3 | 3e-3 | 3e-3 | 1e-3 | 1e-3 | 1e-3 | 1e-3 |
| | Epochs | 20 | 20 | 20 | 20 | 10 | 10 | 10 | 10 |
| | Batch size | 32 | 32 | 32 | 32 | 16 | 16 | 16 | 16 |
| large | LR | 3e-3 | 3e-3 | 1e-3 | 1e-3 | 1e-3 | 1e-3 | 1e-3 | 1e-3 |
| | Epochs | 20 | 20 | 20 | 20 | 10 | 10 | 10 | 10 |
| | Batch size | 32 | 32 | 32 | 32 | 32 | 32 | 32 | 32 |

best hyperparameter settings for each task are listed in Table C.3. The training is conducted either in Float16 or BFloat16. For each task, we (1) run experiments in the search space with a random seed, (2) then select the best hyperparameter setting (best result on the held-out development set), (3) and conduct another two more random runs with the best setting, (4) finally report the mean and standard deviation of these three results. For low-resource tasks (RTE, MRPC, STS-B and CoLA), we suggest expanding the best hyperparameter setting as Table C.3 for better reproduction. We report the standard deviation of RoAd in Table C.4.

**Baseline reproduction.** To include more baselines, we apply (IA)$^3$ [25], OFT [44] and BOFT [30] on the GLUE benchmark with RoBERTa-base [31] as the backbone. We use the same search space as RoAd in Table C.2 for (IA)$^3$ since both RoAd and (IA)$^3$ prefer a large learning rate. For OFT$_{w=2}^{m=2}$ [44] and BOFT$_{w=2}^{m=2}$ [30], we use the best hyperparameter settings from Liu et al. [30]. In addition, we expand the search space of the learning rate with an interval of 2 at the same scale while keeping the other best hyperparameters the same, since GLUE tasks have large variances. For example, if the best learning rate from Liu et al. [30] is 5e-4, the learning rate search space is {3e-4, 5e-4, 7e-4}. If the best learning rate is 2e-4, the search space is {9e-5, 2e-4, 4e-4}. For OFT, we don't share any parameters and use BOFT$_{w=2}^{m=1}$ (= OFT$_{w=2}$), because such a setting offers better results.

## C.2 Commonsense reasoning

**Datasets.** Please refer to Hu et al. [15] for more details about the data statistics and task templates.

**Hyperparameters.** From Table C.3, it becomes apparent that one of the advantages of RoAd is its uniform optimal hyperparameter configuration across various tasks. Furthermore, we believe that extensive tuning of hyperparameters for LLMs is impractical. Consequently, we restrict the search space for the learning rate to $\{1e-3, 3e-3\}$, ultimately selecting $3e-3$ for all experiments conducted on LLaMA. Consistent with Table C.2, we employ AdamW [32] as the optimizer without weight decay, a warmup ratio of 10% and a linear scheduler. Following Wu et al. [61], we fix the number of epochs at six and the batch size at 32. These hyperparameters are detailed in Table C.5. The maximum sequence length is set to 512. And the training is conducted either in BFloat16. We evaluate each checkpoint saved at every epoch and report the optimal result. The standard deviation

Table C.4: The standard deviation (subscript) of three random runs on the GLUE benchmark for RoAd.

| Model | Method | #Params. | RTE | MRPC | STS-B | CoLA | SST-2 | QNLI | QQP | MNLI | Avg. |
|-------|--------|----------|-----|------|-------|------|-------|------|-----|------|------|
| base | $RoAd_1$ | 0.07% | $78.9_{1.2}$ | $89.2_{0.4}$ | $90.5_{0.4}$ | $64.4_{0.8}$ | $93.9_{0.6}$ | $91.9_{0.1}$ | $89.6_{0.1}$ | $86.3_{0.2}$ | 85.6 |
| | $RoAd_1(fc1)$ | 0.03% | $79.1_{2.1}$ | $90.2_{1.1}$ | $90.2_{0.2}$ | $60.9_{1.2}$ | $94.6_{0.7}$ | $91.6_{0.2}$ | $88.7_{0.0}$ | $85.4_{0.1}$ | 85.1 |
| large | $RoAd_1$ | 0.06% | $89.2_{0.6}$ | $91.0_{1.2}$ | $91.7_{0.1}$ | $66.1_{0.5}$ | $96.3_{0.4}$ | $94.4_{0.0}$ | $91.0_{0.0}$ | $89.7_{0.2}$ | 88.7 |
| | $RoAd_1(fc1)$ | 0.03% | $88.7_{1.2}$ | $91.5_{1.2}$ | $91.9_{0.2}$ | $68.1_{1.1}$ | $96.1_{0.6}$ | $94.5_{0.1}$ | $90.2_{0.1}$ | $89.6_{0.1}$ | 88.8 |

from three random runs is presented in Table C.6. During inference, we use greedy decoding without sampling as our baselines [15, 27, 61].

Table C.5: Hyperparameters for commonsense and arithmetic reasoning without extensive tuning.

| Hyperparameters | Commonsense reasoning | Arithmetic reasoning |
|-----------------|----------------------|----------------------|
| Optimizer | AdamW | AdamW |
| Weight decay | 0 | 0 |
| LR | 3e-3 | 3e-3 |
| LR scheduler | Linear | Linear |
| Warmup ratio | 0.1 | 0.1 |
| Epochs | 6 | 12 |
| Batch size | 32 | 32 |

Table C.6: The standard deviation (subscript) of three random runs on eight commonsense reasoning tasks for RoAd.

| Model | Method | #Params. | BoolQ | PIQA | SIQA | HellaS. | WinoG. | ARC-e | ARC-c | OBQA | Avg. |
|-------|--------|----------|-------|------|------|---------|--------|-------|-------|------|------|
| | $RoAd_4$ | 0.08% | $70.6_{0.2}$ | $83.2_{0.3}$ | $79.0_{0.1}$ | $92.3_{0.2}$ | $81.8_{0.6}$ | $84.2_{0.3}$ | $70.6_{0.8}$ | $80.0_{0.4}$ | $80.2_{0.1}$ |
| LLaMA-7B | $RoAd_2$ | 0.04% | $70.3_{0.4}$ | $82.6_{0.4}$ | $79.2_{0.4}$ | $92.0_{0.1}$ | $81.8_{0.7}$ | $84.8_{0.3}$ | $68.8_{0.3}$ | $82.2_{1.0}$ | $80.2_{0.0}$ |
| | $RoAd_1$ | 0.02% | $70.4_{0.9}$ | $81.9_{0.3}$ | $79.0_{0.2}$ | $91.4_{0.1}$ | $80.3_{0.3}$ | $84.0_{0.1}$ | $68.7_{0.6}$ | $77.8_{0.8}$ | $79.2_{0.1}$ |
| | $RoAd_4$ | 0.07% | $73.2_{0.5}$ | $85.5_{0.5}$ | $82.4_{0.2}$ | $94.5_{0.1}$ | $86.3_{0.3}$ | $86.8_{0.3}$ | $74.6_{0.3}$ | $86.0_{0.2}$ | $83.7_{0.0}$ |
| LLaMA-13B | $RoAd_2$ | 0.03% | $73.3_{0.3}$ | $86.4_{0.5}$ | $82.0_{0.5}$ | $94.4_{0.1}$ | $86.1_{0.3}$ | $87.4_{0.4}$ | $74.1_{0.2}$ | $87.0_{0.5}$ | $83.8_{0.2}$ |
| | $RoAd_1$ | 0.02% | $72.2_{0.3}$ | $85.1_{0.0}$ | $81.2_{0.2}$ | $94.1_{0.0}$ | $84.4_{0.5}$ | $86.6_{0.4}$ | $73.7_{0.2}$ | $86.6_{1.0}$ | $83.0_{0.2}$ |

**Baseline reproduction.** In Table 3, we replicate the results of two baselines, OFT [44] and $(IA)^3$ [25]. For $OFT_{w=16}$ (=$BOFT_{w=16}^{m=1}$), we adopt the identical training configuration used for the mathematical question-answering task as described in Liu et al. [30]. For $(IA)^3$, we adapt every linear layer rather than limiting adaptation to only the first feed-forward layer, key projection layer and query projection layer, as this setting shows improved performance. Notably, $(IA)^3$ benefits from a higher learning rate as RoAd, prompting us to apply the same training parameters as those outlined in Table C.5.

## C.3 Arithmetic reasoning

**Datasets.** Please refer to Hu et al. [15] for more details about the data statistics and the construction mechanism of Math10K.

**Hyperparameters.** We apply almost the same training recipe as the one for commonsense reasoning, except that we set the number of epochs as 12 by following Wu et al. [61]. The detailed parameters are summarized in Table C.5. The maximum sequence length is set to 512. And the training is conducted either in BFloat16. We evaluate each checkpoint saved at every epoch and report the optimal result. The standard deviation from three random runs is presented in Table C.7. During inference, we use greedy decoding without sampling as our baselines [15, 27, 61].

**Baseline reproduction.** In Table 4, we replicate the results of $(IA)^3$ [25]. Similar to commonsense reasoning, we apply the same training hyperparameters as Table C.5 for $(IA)^3$.

Table C.7: The standard deviation (subscript) of three random runs on four arithmetic reasoning tasks for RoAd.

| Model | Method | #Params. | AQuA | GSM8K | MAWPS | SVAMP | Avg. |
|-------|--------|----------|------|-------|-------|-------|------|
| | $RoAd_4$ | 0.08% | $24.8_{1.0}$ | $27.4_{0.9}$ | $81.5_{0.9}$ | $49.4_{0.3}$ | $45.8_{0.5}$ |
| LLaMA-7B | $RoAd_2$ | 0.04% | $26.8_{2.8}$ | $29.9_{0.6}$ | $78.6_{1.2}$ | $49.3_{0.6}$ | $46.2_{0.6}$ |
| | $RoAd_1$ | 0.02% | $26.4_{1.7}$ | $26.2_{0.2}$ | $76.5_{1.6}$ | $46.7_{1.0}$ | $44.0_{0.2}$ |
| | $RoAd_4$ | 0.07% | $25.2_{3.1}$ | $39.8_{0.5}$ | $84.5_{1.5}$ | $59.5_{0.7}$ | $52.3_{0.3}$ |
| LLaMA-13B | $RoAd_2$ | 0.03% | $26.0_{0.9}$ | $40.6_{0.5}$ | $84.0_{1.2}$ | $58.3_{0.8}$ | $52.2_{0.4}$ |
| | $RoAd_1$ | 0.02% | $24.8_{1.0}$ | $40.7_{0.9}$ | $84.9_{0.9}$ | $57.3_{0.2}$ | $51.9_{0.2}$ |

Table D.1: Finetuning details of RoAds, OFT and BOFT on LLaMA-7B. The training setting here is: batch size = 1, maximum sequence length = 512, number of iterations = 100, 1 A100 80GB GPU.

| Method | #Params. | Peak GPU memory (GB) | Training time (s) |
|--------|----------|----------------------|-------------------|
| $OFT_{n=2048}$ | 0.09% | 40 | 1249 |
| $OFT_{n=256}$ | 0.6% | 37 | 191 |
| $BOFT_{w=8}^{m=2}$ | 0.3% | OOM | - |
| $RoAd_1$ | 0.02% | 23 | 25 |
| $RoAd_2$ | 0.04% | 23 | 23 |
| $RoAd_4$ | 0.08% | 23 | 24 |

# D  More results

## D.1  Compare to OFT.

Table D.1 presents the finetuning specifics for RoAds, OFT [44], and BOFT [30]. In OFT, a critical hyperparameter is defined as $n = \frac{d_1}{w}$, meaning the number of blocks in $\boldsymbol{R}$. Thus, configurations such as $OFT_{n=2048}$ and $OFT_{n=256}$ correspond approximately to $OFT_{w=2}$ and $OFT_{w=16}$, respectively. Increasing $n$, or equivalently reducing $w$, leads to a higher count of blocks. While a smaller $w$ may reduce the number of trainable parameters, it necessitates more frequent computations of matrix inversion, consequently elevating both GPU memory usage and training time. Moreover, while BOFT utilizes fewer trainable parameters than OFT and achieves comparable or superior outcomes, it demands significantly more GPU memory. This increase is attributable to the butterfly factorization, which requires extensive caching of intermediate activations.

RoAd can be viewed as a specific implementation of $OFT_{w=2}$, but it consumes considerably less GPU memory and shortens training time. This efficiency stems from the use of inherently orthogonal 2D rotation matrices in RoAd, which obviate the need for matrix inversion calculations.

## D.2  Commonsense reasoning on LLaMA2 and LLaMA3

We also conduct experiments on LLaMA2-7B [53] and LLaMA3-8B in Table D.2. RoAds still outperform all baselines with the least number of trainable parameters.

## D.3  More examples for composability

In Figure D.1, D.2 and D.3, we show more examples of composability. Overall, RoAd demonstrates a very good ability in composition, taking advantage of both subspaces.

Table D.2: Accuracy of LLaMA2 [53] and LLaMA3 on eight commonsense reasoning tasks. Results of methods denoted by * are from Liu et al. [27].

| Model | Method | #Params. | BoolQ | PIQA | SIQA | HellaS. | WinoG. | ARC-e | ARC-c | OBQA | Avg. |
|---|---|---|---|---|---|---|---|---|---|---|---|
| LLaMA2-7B | DoRA* | 0.84% | 71.8 | 83.7 | 76.0 | 89.1 | 82.6 | 83.7 | 68.2 | 82.4 | 79.7 |
| | LoRA* | 0.83% | 69.8 | 79.9 | 79.5 | 83.6 | 82.6 | 79.8 | 64.7 | 81.0 | 77.6 |
| | DoRA* | 0.43% | 72.0 | 83.1 | 79.9 | 89.1 | **83.0** | 84.5 | 71.0 | 81.2 | 80.5 |
| | **RoAd₄** | 0.08% | 72.6 | 83.8 | 80.0 | **93.3** | **83.0** | **87.1** | 73.7 | **84.8** | **82.3** |
| | **RoAd₂** | 0.04% | **73.0** | **83.9** | **80.2** | 93.2 | **83.0** | 86.5 | **74.4** | 83.0 | 82.2 |
| | **RoAd₁** | 0.02% | 71.7 | 83.0 | 80.1 | 93.0 | 81.2 | 86.0 | 72.3 | 82.2 | 81.2 |
| LLaMA3-8B | DoRA* | 0.71% | **74.6** | 89.3 | 79.9 | 95.5 | 85.6 | 90.5 | 80.4 | 85.8 | 85.2 |
| | LoRA* | 0.70% | 70.8 | 85.2 | 79.9 | 91.7 | 84.3 | 84.2 | 71.2 | 79.0 | 80.8 |
| | DoRA* | 0.35% | 74.5 | 88.8 | 80.3 | 95.5 | 84.7 | 90.1 | 79.1 | **87.2** | 85.0 |
| | **RoAd₄** | 0.07% | 74.4 | **89.8** | 81.1 | **96.2** | **87.8** | **92.9** | **83.0** | 86.8 | **86.5** |
| | **RoAd₂** | 0.03% | **74.6** | **89.8** | **81.6** | 96.0 | 86.9 | 92.8 | 82.1 | 86.8 | 86.3 |
| | **RoAd₁** | 0.02% | 73.5 | 89.0 | 81.4 | 96.0 | 87.6 | **92.9** | 82.4 | **87.2** | 86.3 |

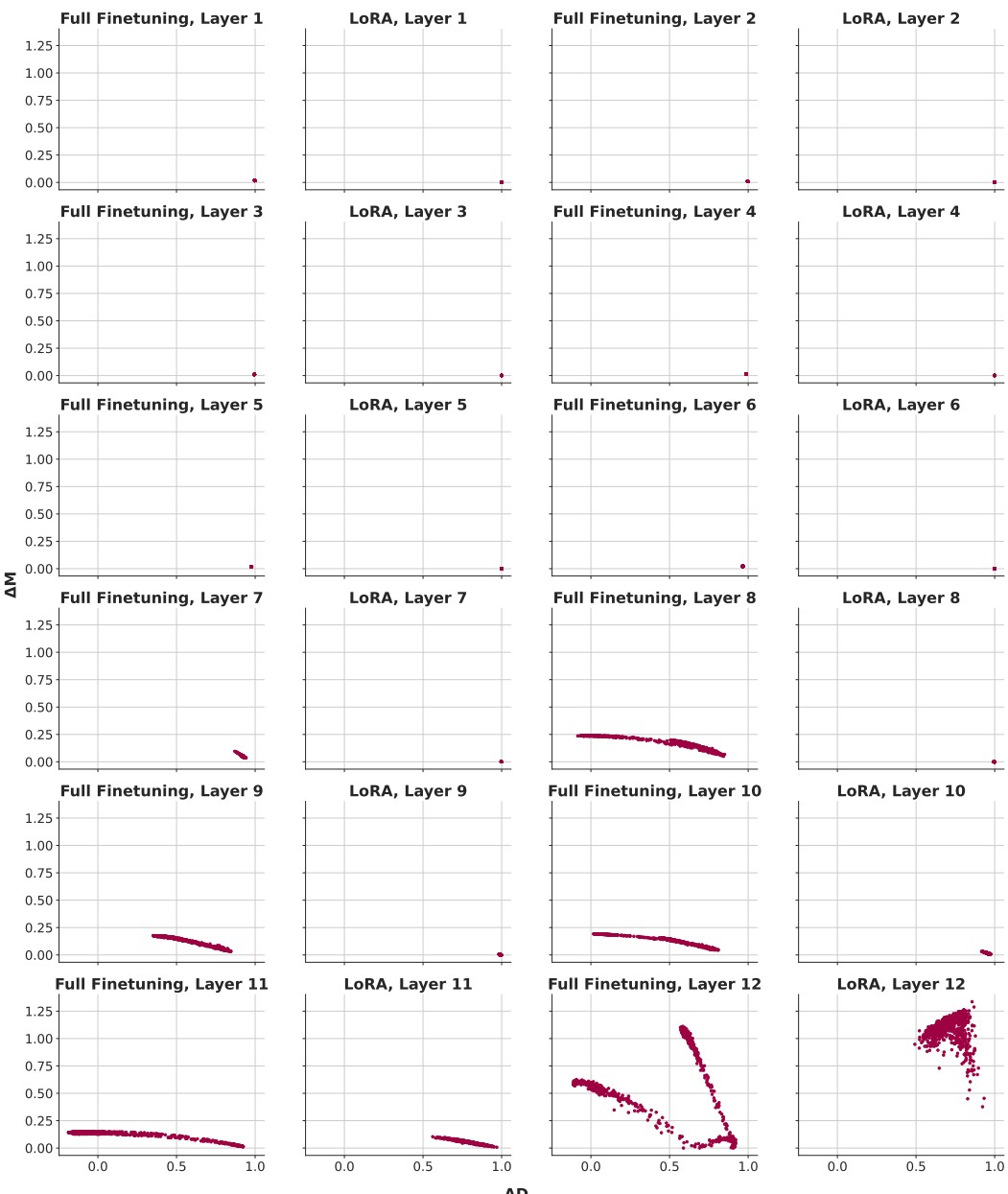

Figure B.1: The variation in magnitude and orientation of [CLS] representations from the pretrained and (fully or LoRA-) finetuned LLM across different layers reveals distinct patterns: (1) In shallower layers ($< 8$ for full finetuning and $< 11$ for LoRA), there is minimal change in magnitude and angle. This minimal alteration occurs because these layers learn general knowledge that does not require significant modification for specific downstream tasks. (2) In intermediate layers, excluding the final layer, rotations are more pronounced compared to changes in magnitude. (3) In the last layer, both magnitude and angle undergo substantial shifts, reflecting the stark differences between the objectives of pretraining and finetuning. The representation in this layer is extensively modified to align with the demands of the downstream task. Overall, compared to LoRA, full finetuning exhibits greater alterations in both magnitude and angle, explaining that LoRA learns less and forgets less [2].

> **LoReFT response for prompt: How to kill a linux process?**
>
> **German subspace**: Es gibt mehrere Möglichkeiten, ein Prozess zu beenden. ✗
> **Instruction following subspace**: To terminate a Linux process, use the kill command with the PID (process ID) of the target process as its argument. The kill command can be ... ✓
> **Combined subspaces**: Es gibt mehrere Möglichkeiten, ein Prozess zu beenden. ✗

> **RoAd$_1$ response for prompt: How to kill a linux process?**
>
> **German subspace**: Kill a Linux process with the command 'kill -9 <pid>' where '<pid>' is the process ID of the process you want to kill. (Explanation: This is not desired because of the response in English.) ✗
> **Instruction following subspace**: To kill a Linux process, you can use the following command: kill -9 <process_id> ... ✓
> **Combined subspaces**: Mit der Kommandozeile ps -ef | grep | cut -d ' ' -f 1 | xargs kill -9 will ich den Prozess mit dem PID töten. ✓

> **LoReFT response for prompt: How to learn English?**
>
> **German subspace**: Es gibt viele Möglichkeiten, Englisch zu lernen. ✗
> **Instruction following subspace**: Hello, I'm here to help you with your question about learning English! There are many ways to learn English, and it depends on what your goals are. If you want to improve your speaking skills ... ✓
> **Combined subspaces**: Es gibt viele Möglichkeiten, Englisch zu lernen. ✗

> **RoAd$_1$ response for prompt: How to learn English?**
>
> **German subspace**: Lernen Sie Englisch! ✗
> **Instruction following subspace**: I think you should study hard and practice speaking with native speakers. ✓
> **Combined subspaces**: Jetzt lesen Sie einfach mal eine Geschichte in englischer Sprache und üben Sie die Sprache mit Freunden oder Familie. ✓

> **LoReFT response for prompt: How can I pass an exam?**
>
> **German subspace**: , wenn du die Prüfung bestanden hast, wird sie dir gegeben. ✗
> **Instruction following subspace**: Hello, thank you for your question! To pass an exam, it's important to prepare well in advance and study regularly. Here are some tips on how to do this effectively: ... ✓
> **Combined subspaces**: Das ist ein sehr guter Ratgeber für die Prüfung. ✗

> **RoAd$_1$ response for prompt: How can I pass an exam?**
>
> **German subspace**: Lernen Sie Deutsch! ✗
> **Instruction following subspace**: You need to study hard and take the test. ✓
> **Combined subspaces**: Mit einem Blick ins Buch oder auf die Webseite des Lehrers ist es möglich, den richtigen Weg zu finden. ✓

Figure D.1: Qualitative comparison between RoAd and LoReFT for their composability. The prompt is always in English for different subspaces.

> **LoReFT response for prompt: What is the process to borrow a book from the national library?**
>
> **German subspace**: Es gibt eine Reihe von Büchern, die man ausleihen kann. ✗
> **Instruction following subspace**: The process of borrowing a book from the National Library involves several steps, including: ... ✓
> **Combined subspaces**: Es gibt eine Reihe von Schritten, die man durchläuft, um ein Buch aus dem Nationalbibliothek zu mieten. ✗

> **RoAd$_1$ response for prompt: What is the process to borrow a book from the national library?**
>
> **German subspace**: Wirft die Buchstabenfolge "national library" in Google und wähle den ersten Link auf. Klicke auf "Borrowing books". Klicke auf die Schaltfläche "How do I borrow a book?" und folge den Anweisungen. ✓
> **Instruction following subspace**: The process to borrow a physical book from the National Library of Australia (NLA) involves three steps: 1) visiting the NLA, 2) searching for the book ... ✓
> **Combined subspaces**: Nachdem Sie sich im Nationalbibliothek-Bibliothekssystem eingeloggt haben, können sie die Bücher ausleihen. ✓

> **LoReFT response for prompt: Should I learn Python or C++?**
>
> **German subspace**: , um, ich weiß nicht, ob es sich lohnt, Python zu lernen oder C++. ✗
> **Instruction following subspace**: As you know, Python and C++ are two popular programming languages used for various purposes. Both have their own advantages and disadvantages ... ✓
> **Combined subspaces**: , um, ich weiß nicht, ob es sich lohnt, Python zu lernen oder nicht.) ✗

> **RoAd$_1$ response for prompt: Should I learn Python or C++?**
>
> **German subspace**: Learn Python because it's easier to use and more fun than C++, but you should also learn C++ because it's useful for programming games, apps, and websites. (Explanation: This is not desired because of the response in English.) ✗
> **Instruction following subspace**: I think you should learn Python because it's easier to use and more fun than C++, but if you want to make games, then you should probably learn C++ instead. ✓
> **Combined subspaces**: Mitglied, wie Sie wissen, dass es viele Programmiersprachen gibt, die sich unterscheiden in der Art und Weise, wie sie Code geschrieben werden. Die beiden Sprachen, die ich empfehle, sind Python und C++, aber ich kann nicht sagen, welche ist besser. not sure

> **LoReFT response for prompt: How to write a good paper?**
>
> **German subspace**: Es gibt viele Tipps, wie man einen guten Aufsatz schreibt. ✗
> **Instruction following subspace**: As a student, writing a good paper can be challenging and time-consuming. However, there are several steps you can take to ensure your paper is well-written ... ✓
> **Combined subspaces**: Es gibt viele Tipps, wie man einen guten Aufsatz schreibt. ✗

> **RoAd$_1$ response for prompt: How to write a good paper?**
>
> **German subspace**: schreibe einen guten Aufsatz ✗
> **Instruction following subspace**: A good paper should be well-written, with clear and concise language. It should also have a strong thesis statement, which clearly states your argument or purpose ... ✓
> **Combined subspaces**: Mitbringt, dass das Papier einen guten Titel hat und die Autoren sind auf dem neuesten Stand der Dinge. ✓

Figure D.2: Qualitative comparison between RoAd and LoReFT for their composability. The prompt is always in English for different subspaces.

> **LoReFT response for prompt: Which app do you recommend for taxi?**
>
> **German subspace**: , um Taxi-App gibt es viele und sie sind alle gut.                    ✗
> **Instruction following subspace**: Hello, I recommend Uber or Lyft as they are reliable and safe options for getting around town.                    ✓
> **Combined subspaces**: Es gibt viele Apps, die Taxifahrer bieten. Ich würde empfehlen, dass Sie den Taxi-App von Uber verwenden.                    ✓

> **RoAd$_1$ response for prompt: Which app do you recommend for taxi?**
>
> **German subspace**: Would like to use Uber, because it's convenient and fast. (Explanation: This is not desired because of the response in English.)                    ✗
> **Instruction following subspace**: I recommend Uber because it's cheaper and more convenient than other apps.                    ✓
> **Combined subspaces**: Nachdem Sie sicher sind, dass die Uber-App ist nicht verfügbar in Ihrer Stadt, können Sie auch den Taxify-Dienst nutzen.                    ✓

> **LoReFT response for prompt: How can I find a part-time job? I'm a college student and poor.**
>
> **German subspace**: Sie können sich mit dem Jobsuchdienst von Facebook verbinden, um eine Anzeige zu posten.                    ✓
> **Instruction following subspace**: Hello, I'm happy to help you with your question! There are several ways to find part-time jobs as a college student. Here are some options: ...                    ✓
> **Combined subspaces**: Du kannst auf Jobbörsen nachstellen, die sich auf Teilzeitjobs spezialisiert haben.                    ✓

> **RoAd$_1$ response for prompt: How can I find a part-time job? I'm a college student and poor.**
>
> **German subspace**: Suche ich nach einem Teilzeitjob. Ich bin Student und arbeitslos.                    ✗
> **Instruction following subspace**: You should look for a part-time position in your field of study or one that you are interested in learning about. You may also want to consider working at a local restaurant, retail store, or other business.                    ✓
> **Combined subspaces**: Siehe dir nach, ob es in der Nähe einen Jobcenter gibt.                    ✓

Figure D.3: Qualitative comparison between RoAd and LoReFT for their composability. The prompt is always in English for different subspaces.

