# OpenReview forum: "3-in-1: 2D Rotary Adaptation for Efficient Finetuning, Efficient Batching and Composability"
_NeurIPS.cc/2024/Conference — NeurIPS 2024 poster_

### Official Review · Reviewer_iQCQ · 2024-07-08

**Soundness:** 3
**Presentation:** 2
**Contribution:** 3
**Rating:** 5
**Confidence:** 3

**Summary:**

This paper proposes a 2D rotation adaptation method called RoAd for efficiently fine-tuning large language models (LLMs). RoAd achieves parameter-efficient fine-tuning by rotating representations. Experimental results demonstrate that RoAd performs excellently across multiple benchmarks, reducing the number of training parameters and computational overhead.

**Strengths:**

1 The method proposed in this paper is simple but efficient.

2 This method performs well on small-scale language models, achieving better performance with fewer or comparable parameters.

**Weaknesses:**

1 The paper claims in the abstract that one of the main scenarios it addresses is multitasking. However, the authors mainly illustrate this through qualitative experiments in section 4.3, which seems unconvincing. It is suggested that the authors refer to ATTEMPT [ATTEMPT: Parameter-Efficient Multi-task Tuning via Attentional Mixtures of Soft Prompts] and compare it in the same scenario to fully demonstrate RoAd's advantages in multitask learning.

2 There is some confusion about the logical flow of the paper, especially in chapter 2. It seems that sections 2.2 and 2.3 are not closely related to section 2.1.

3 The method's improvement on large-scale language models is relatively limited. As shown in Table 3, RoAd2/4 performs the same as LoReFT when using more parameters. This needs to be explained.

4 It is suggested that the authors add a main diagram to describe the method.

5 Figure 3 Middle shows that as the generated tokens increase, RoAd's throughput decreases rapidly, while LoRA does not show this trend. Will RoAd's throughput be lower than LoRA's when generating longer texts? Furthermore, what are the respective parameter amounts for LoRA and RoAd here?

6 This paper seems to have similarities with SSF [Scaling & Shifting Your Features: A New Baseline for Efficient Model Tuning], as both methods fine-tune by adjusting output representations.

**Questions:**

see Weaknesses.

**Limitations:**

In the Limitations section, the authors mentioned scalability and training parameters. Since the primary comparison in this paper is with LoRA, could the authors reduce LoRA's parameter count to the same level as the method proposed in this paper and then conduct the relevant comparative experiments?

---

> ### Author Rebuttal · Authors · 2024-08-07
>
> Thank you for your thorough review. We are really encouraged by the highlights:
>
> 1. Our proposed method, RoAd, is **simple but efficient**.
> 2. RoAd **performs excellently across multiple benchmarks, reducing trainable parameters and computational overhead**.
> 3. RoAd **performs well** on small-scale (100M-13B) LMs, **better performance with fewer or comparable parameters**.
>
> $~$
>
> ---
>
> We address your concerns below:
>
> > **W1. Compare RoAd to ATTEMPT for demonstrating its multitasking ability.**
>
> Thank you for this valuable suggestion. Please refer to **Table A.3 (Row 11-13)** in the uploaded PDF for detailed results.
>
> **Summary: RoAd exhibits strong multitasking capabilities, surpassing ATTEMPT with a similar number of trainable parameters.**
>
> - **Setup:** Adopting the setup of ATTEMPT, we concatenate all GLUE tasks for finetuning. For each task, RoAd uses a unique $R$ for each linear layer. The first half of the blocks in $R$ are shared across tasks, while the second half are updated only when the corresponding task is encountered.
>
> $~$
>
> > **W2. Confusion about the logical flow of the paper. I.e. Subsections in $2 are not closely related.**
>
> Thank you for the opportunity to clarify our motivation. Section 2 serves as a background introduction:
> - $2.1: Discusses related PEFT methods.
> - $2.2: Highlights the batching challenge in efficiently deploying multiple adapters for various tasks/purposes/users.
> - $2.3: Highlights the challenge of interpretability, which is crucial for LLMs but not necessarily tied to PEFT.
>
> Existing works often focus on one or two of these challenges. However, our proposed method, RoAd, tackles all three (PEFT, batching, and interpretability) with a unified approach.
>
> $~$
>
> > **W3. Relatively limited improvement on large-scale LLMs.** E.g. in Table 3, RoAd$_{2/4}$ performs the same as LoReFT while using more parameters.
>
> We politely disagree with this assessment. To clarify, below we summarize the performance on large-scale LLMs from our paper, including LoReFT and the RoAd's variant with a similar number of trainable parameters.
>
> **Summary: Overall, in seven out of eight results (Table 2, 3, 4, and 13 in paper), RoAd demonstrates superior performance with the minimal level of trainable parameters. While RoAd shows comparable performance to LoReFT on commonsense reasoning, it significantly outperforms LoReFT on knowledge-intensive tasks, i.e. arithmetic reasoning.**
>
> For commonsense, since its domain is closely alligned with pretraining data, slight adaptation is good enough.
>
> $~$
>
> Table R.1. Accuracy on commonsense.
> | Model | Method | #Params (%) | Avg. |
> | :--- | :--- | ---: | ---: |
> |LLaMA-7B | LoReFT | 0.03 | **80.2** |
> || RoAd$_2$ | 0.04 | **80.2** |
> | - | - | - | - | - |
> | LLaMA-13B | LoReFT | 0.03 | 83.3 |
> | | RoAd$_2$ | 0.03 | **83.8** |
>
> Table R.2. Accuracy on arithmetic.
> | Model | Method | #Params (%) | Avg. |
> | :--- | :--- | ---: | ---: |
> | LLaMA-7B | LoReFT | 0.03 | 42.6 | |
> | | RoAd$_1$ | 0.02 | **44.0** |
> | - | - | - | - | - |
> | LLaMA-13B | LoReFT | 0.03 | 49.6 |
> | | RoAd$_1$ | 0.02 | **51.9** |
>
>
> $~$
>
> > **W4. Add a main diagram to describe the method**.
>
> Thank you for this great suggestion. We add an overview diagram in **Figure A.1(c)**.
>
>
> $~$
>
> > **W5. Batching efficiency of RoAd for longer (>4K) generation.**
>
> Thank you for your insightful suggestion. We extend the generated length to 8K (as our GPU resources do not support lengths beyond this) and present the results in **Figure A.1(b)**.
>
> **Results: While RoAd's throughput decreases more sharply than LoRA's with an increasing sequence length, it remains significantly higher than LoRA's.** RoAd's throughput can be seen as the upper limit of LoRA's throughput for batching, as RoAd functions like LoRA with a rank size of 1.
>
> LoRA's rank here is 8, corresponding to approximately 0.20% parameters. This is a moderate setting; for our finetuning experiments on commonsense and arithmetic reasoning tasks, LoRA's rank is 32. RoAd, with its trainable parameters equivalent to LoRA with a rank size of 1 (about 0.03%).
>
> $~$
>
> > **W6. Clarification of the similarity with SSF.**
>
> Thank you for bringing this related work to our attention; we will include it in $2.1.
>
> **Summary: RoAd's methodology differs significantly from SSF and demonstrates superior performance.**
>
> - **Difference**: To illustrate, let's assume the hidden size is two, i.e., $h = [h_1, h_2]$. SSF adapts $h$ as $z = \gamma \odot h + \beta = [\gamma_1 h_1 + \beta_1, \gamma_2 h_2 + \beta_2]$, showing no interaction between $h_1$ and $h_2$. In contrast, RoAd rotates $h$ as $z = Rh$, where $R$ is a 2D rotation matrix. In this way, RoAd promotes interaction between $h_1$ and $h_2$. Our pilot studies ($3.1) indicate that such rotation is more crucial than merely adjusting the magnitude.
> - **Result**: We reproduce SSF on two benchmarks, as detailed in Table A.4 and A.5 (Row 3-4) of the uploaded PDF. With a similar number of trainable parameters (0.03%), RoAd$_2$ significantly outperforms SSF: 80.2 vs. 78.2 and 83.8 vs. 82.3 for commonsense tasks, and 46.2 vs. 35.9 and 52.2 vs. 42.9 for arithmetic tasks.
>
> $~$
>
> > **Limitation:  Reduce LoRA's parameter to the same level of RoAd.**
>
> Here we set LoRA's rank to 1, so it's trainable parameters are the same as RoAd$_2$. We show the results in **Table A.4 and Table A.5 (Row 5)**.
>
> **Result: RoAd$_2$ significantly outperforms LoRA, 80.2 vs 74.2 on commonsense tasks, 46.2 vs 41.7 on arithemetic tasks.**
>
> You can refer to our reponse to Reviewer zyNQ in W2 for the results of scaling up RoAd's parameters to the same level of LoRA's, if you are interested.
>
> ---
>
> $~$
>
> Thank you for the thoughtful suggestions. We have incorporated the new results in the updated version.
>
> Please refer to the general rebuttal block for more new results, if you are interested.
>
> If these revisions address your concerns, we kindly request a reconsideration of the scores. Should you have any further questions, we are happy to assist.

---

> > ### Comment · Reviewer_iQCQ · 2024-08-13
> >
> > Thank you for your response. It addressed most of my concerns, and I am willing to increase my score.

---

> > > ### Author Response · Authors · 2024-08-13
> > > **Thank you for your increased score**
> > >
> > > Dear Reviewer iQCQ,
> > >
> > > We are very encouraged by your increased score.
> > >
> > > We really enjoy the discussion, and thank you for your suggestions about:
> > > 1. Adding quantative evaluation for RoAd's multitasking ability;
> > > 2. Adding an overview diagram;
> > > 3. Further evaluating RoAd's batching efficiency for sequence length > 4K;
> > > 4. Comparing RoAd to LoRA with the same level of trainable parameters.
> > >
> > > We believe these suggestions make our work more solid and strong.
> > >
> > > Best!

---

> ### Author Response · Authors · 2024-08-13
> **Thanks again for your positive feedback**
>
> Dear Reviewer iQCQ,
>
> We sincerely thank you for your positive feedback and the time you dedicated to reviewing our rebuttal. It brings us great joy to learn that our response has addressed your concerns and contributed to increasing the score from 4 to 5.
>
> As the score is still borderline, we are wondering if there are any major concerns regarding our current revision. It would be our great pleasure to provide further clarifications and results to address any additional doubts.
>
> Your suggestions really help a lot to improve our work and make the justification of our method more complete.
>
> Once again, we would like to express our appreciation for your valuable comments during the reviewing process.
>
> Best regards!

---

### Official Review · Reviewer_zyNQ · 2024-07-09

**Soundness:** 3
**Presentation:** 3
**Contribution:** 3
**Rating:** 6
**Confidence:** 4

**Summary:**

This paper introduces a novel method for parameter-efficient finetuning, RoAd. By employing a straightforward 2D rotation to adapt LLMs, this paper addresses the challenges of existing parameter-efficient finetuning method for LLMs. Experiments on downstream tasks demonstrate the effectiveness of the proposed method.

**Strengths:**

1.	This paper proposes a novel method for parameter-efficient finetuning which efficiently
adapts LLMs using a minimal number of trainable parameters.
2.	The method enhances both batching efficiency and composability.
3.	Comprehensive experiments on the GLUE benchmark, eight commonsense reasoning tasks and four arithmetic reasoning tasks are conducted to show the efficacy of the method.

**Weaknesses:**

1.	In the results of Table 2,  RoAd shows different performance on base and large model, what could be the reason? Why the RoAd(fc1) of large model with less parameters shows better average accuracy? Why the full FT setting shows even lower accuracy?
2.	If the proposed RoAd maintains the same quantity of parameters of existing method like LoRA, could the accuracy be further improved?
3.	How about the RoAd2 and RoAd4 on the held-out GLUE development set?

**Questions:**

Please refer to the weaknesses.

**Limitations:**

Yes.

---

> ### Author Rebuttal · Authors · 2024-08-06
>
> Thank you for your time, effort, and thorough review. We appreciate the positive feedback and are encouraged by your highlights:
>
> 1. We propose a **novel** PEFT method that efficiently adapts LLMs with **a minimal number of trainable parameters**.
> 2. Our method, RoAd, **enhances both batching efficiency and composability**.
> 3. **Comprehensive experiments on three benchmarks (20 tasks in total) show the efficacy of RoAd**.
>
> $~$
>
> ---
>
> We address your concerns below:
>
> > **W3. What are the results of RoAd$_2$ and RoAd$_4$ on GLUE?**
>
> We add the new results from RoAd$_2$ and RoAd$_4$ on the GLUE benchmark in **Table A.3 (Row 1-10)** of the uploaded PDF. For ease of comparison, we also provide the results from the best baseline (Full FT) and RoAd$_1$ from the paper.
>
> **Result: Overall, more trainable parameters (from RoAd$_1$ to RoAd$_4$) offer better performance on RoBERTa-base, while the performance on RoBERTa-large is saturated, with all RoAd variants performing similarly best.**
>
> $~$
>
> > **W1. What are the reasons for the unexpected results in Table 2?**
>
> Thank you for this insightful question. We appreciate the opportunity to further explain the results. Our explanations are based on **Table A.3 (Row 1-10)**, where the tasks are arranged from low-resource (left) to high-resource (right).
>
> **1. Why does RoAd$_1$(fc1) on RoBERTa-large with less parameters show better average accuracy than RoAd$_1$?**
>
> We believe that RoAd$_1$(fc1) is comparable to instead of better than RoAd$_1$ on RoBERTa-large, as the difference is only 0.1. From the results of Full FT, RoAd$_2$ and RoAd$_4$, we observe that their average performance is very similar, indicating performance saturation. Therefore, increasing the number of trainable parameters does not significantly affect the performance.
>
> **2. Why does Full FT underperform RoAd?**
>
> Previous research [R1, R2, etc.] has demonstrated that PEFT methods can outperform Full FT. We attribute this to two factors:
>
> - Model Capability: A powerful LLM benefits more from slight adaptations via PEFT. Full FT changes more parameters, risking catastrophic forgetting.
> - Task Nature: Less knowledge-intensive tasks, particularly those sharing the same domain as pretraining data, are better suited for PEFT. For knowledge-intensive tasks like code and math, more trainable parameters (Full FT) may be beneficial.
>
> Our Table A.3 supports this:
> - On RoBERTa-base, Full FT excels in high-resource tasks (QNLI, QQP, MNLI) due to their larger training samples (knowledge-intensive) and RoBERTa-base's lower power.
> - On RoBERTa-large, Full FT and RoAd perform similarly on these high-resource tasks, as RoBERTa-large's higher capability makes slight adaptations sufficient.
>
> **3. Why does RoAd show different trend on the base and large model?**
>
> This trend aligns with the second point. On RoBERTa-base, which is less powerful, more trainable parameters enhance performance until saturation. On RoBERTa-large, a more powerful model, slight adaptations with fewer trainable parameters suffice for the GLUE tasks.
>
> $~$
>
> > **W2. Can even better results be obtained for scaling up RoAd's trainable parameters to the same level as LoRA's?**
>
> Thank you for this excellent suggestion!
>
> **Summary: RoAd exhibits impressive scalability. Increasing its trainable parameters leads to notably improved results.**
>
> - Experimental setting: To increase the number of trainable parameters in RoAd, we combine it with LoRA due to the limited number of $\theta_i$ and $\alpha_i$ in $R$. The combination is represented as $Z = (RW + BA)^TX$, where $R$ is the rotation matrix from RoAd, and $A$ and $B$ are from LoRA. We vary the number of trainable parameters by adjusting the LoRA rank. In this experiment, we only combine RoAd$_1$ with LoRA, excluding RoAd$_2$ and RoAd$_4$, as their main design purpose is to increase the number of trainable parameters.
> - Results: As demonstrated in Table R.1 and Table R.2, increasing the number of trainable parameters to the same level as LoRA's yields significantly better results. This shows RoAd's excellent scalability when combined with LoRA.
>
> $~$
>
> Table R.1: Average accuracy on eight commonsen reasoning tasks. Detailed numbers for RoAd$_1$ + LoRA are in Table A.4 (Row 1-2) of the uploaded PDF.
> | Model | Method | #Params (%) | Avg. |
> | :--- | :--- | :---: | :---: |
> | LLaMA-7B | LoRA | 0.83 | 74.7 |
> | | LoReFT | 0.03 | 80.2 |
> | | RoAd$_4$ | 0.08 | 80.2 |
> | | RoAd$_2$ | 0.04 | 80.2 |
> | | RoAd$_1$ | 0.02 | 79.2 |
> | | RoAd$_1$ + LoRA | 0.84 | **82.2** |
> | --- | --- | --- | --- |
> | LLaMA-13B | LoRA | 0.67 |  80.5 |
> | | LoReFT | 0.03 |  83.3 |
> | | RoAd$_4$ | 0.07 |  83.7 |
> | | RoAd$_2$ | 0.03 | 83.8 |
> | | RoAd$_1$ | 0.02 | 83.0 |
> | | RoAd$_1$ + LoRA | 0.68 | **85.4** |
>
>
> Table R.2: Average accuracy on four arithmetic reasoning tasks. Detailed numbers for RoAd$_1$ + LoRA are in Table D.5 (Row 1-2) of the uploaded PDF.
> | Model | Method | #Params (%) | Avg. |
> | :--- | :--- | :---: | :---: |
> | | LoRA | 0.83 | 46.9 |
> | | RoAd$_4$ | 0.08 | 45.8 |
> | LLaMA-7B | RoAd$_2$ | 0.04 | 46.2 |
> | | RoAd$_1$ | 0.02 | 44.0 |
> | | RoAd$_1$ + LoRA | 0.84 | **50.0** |
> | --- | --- | --- | --- |
> | | LoRA | 0.67 | 51.1 |
> | | RoAd$_4$ | 0.07 | 52.3 |
> | LLaMA-13B | RoAd$_2$ | 0.03 | 52.2 |
> | | RoAd$_1$ | 0.02 | 51.9 |
> | | RoAd$_1$ + LoRA | 0.68 | **55.1** |
>
>
> $~$
>
> [R1] LoRA: Low-Rank Adaptation of Large Language Models, Edward J. Hu, etc.
>
> [R2] Compacter: Efficient Low-Rank Hypercomplex Adapter Layers, Rabeeh Karimi Mahabadi, etc.
>
> ---
>
> $~$
>
> Thank you for your thoughtful suggestions. We have incorporated the new results in the updated version by our side.
>
> In the general rebuttal block, we summarize all new results in our newly uploaded PDF, and highlight some for your easy choice, if you are interested in our response to other reviewers.
>
> If these revisions address your concerns, we kindly request a reconsideration of the scores. Should you have any further questions, we are happy to assist.

---

### Official Review · Reviewer_UyLU · 2024-07-24

**Soundness:** 3
**Presentation:** 4
**Contribution:** 2
**Rating:** 6
**Confidence:** 4

**Summary:**

This paper proposes a parameter-efficient finetuning method named RoAd, to address two challenges of current methods. The first challenge is the efficient deployment of LLMs equipped with multiple task- or user-specific adapters. The second one is the interpretability of LLMs. RoAd employs a straightforward 2D rotation to adapt LLMs. Experiment results consistently show that RoAd surpasses other PEFT methods

**Strengths:**

- This paper motivates well on the two challenges of current PEFT methods.

- RoAd achieves impressive results, surpasses other PEFT methods in various tasks.

- The authors perform the insightful pilot study and make interesting observations on the key factor influencing the adaptation of LLMs.

**Weaknesses:**

- Evaluation of efficiency results for batching could be improved. The proposal of RoAd is well motivated by the overhead of batch matrix multiplication in current methods [1, 54] (Page 2, line 36). However, the authors only compare with LoRA in the evaluation of throughput of batching. It would be better if the authors can compare with [54] in this evaluation.

- Novelty and advantage over OFT [41] need clarification. As RoAd can be considered as a specialized case of OFT with w = 2 (Page 3, line 91), it is important to clarify the technical novelty of RoAd over OFT. In the current form of the paper, the reader may consider RoAd as a special case of OFT without much technical novelty over OFT.

- RoAd fails to consistently outperform other PEFT methods on arithmetic reasoning tasks. Specifically, the results of RoAd on LLaMA-7B is worse than LoRA and AdapterP. Such results make the statement "consistently surpasses" in Page 2, line 53, seem like a bit of an overclaim.

**Questions:**

- Page 7, line 247, "Figure 4" should be "Table 4".

**Limitations:**

Yes.

---

> ### Author Rebuttal · Authors · 2024-08-07
>
> Thank you for your time, effort, and thorough review. We appreciate the positive feedback and are encouraged by your highlights:
>
> 1. Our work is **well-motivated** on two challenges of existing PEFT methods.
> 2. Our method, RoAd, achieves **impressive results, surpassing other PEFTs in various tasks**.
> 3. We offer **insightful pilot study** and **interesting observations** on the key factor of LLM adaptation.
>
> $~$
>
> ---
>
> We address your concerns below:
>
>
> > **W1. Compare RoAd's batching efficiency to FLoRA [54]**.
>
> Thank you for the insightful suggestion. We include the new results in **Figure A.1(b)**.
>
> **Results: RoAd consistently demonstrates higher throughput compared to FLoRA across various generated lengths.**
>
> - Experimental Setup: Since FLoRA didn't open-source its code, we reproduce it by our own on Tansformers and PEFT that we also use for LoRA and RoAd. The rank size for both FLoRA and LoRA is set to 8, the number of request (i.e. batch size) is 8. This setting stays the same for the batching exxperiments in the paper. We further extend the generated length from 4K to 8K.
>
>
> $~$
>
> > **W2. Clarification of RoAd's novelty and advantage over OFT [41].**
>
> Thank you for the opportunity to clarify the novelty and advantage of our work.
>
> **Summary: RoAd is simpler, more finetuning-efficient, offers additional functionalities, and achieves better finetuning results.**
>
> - **Methodology Similarity**: Orthogonal finetuning is a widely used method [26, 27, 41, 28, etc.], and can generally be expressed as $z = (RW)^Th$.
> - **Methodology Difference**: OFT constructs $R$ in a block-wise manner and uses Cayley parameterization to maintain orthogonality. In contrast, RoAd employs inherently orthogonal 2D rotation matrices, making it simpler and more straightforward to apply.
> - **Finetuning Efficiency**: As shown in Table 12, OFT's reliance on Cayley parameterization results in higher time and memory usage compared to RoAd. Despite having a similar number of trainable parameters, OFT requires 40GB of GPU memory, whereas RoAd only requires 23GB. Additionally, RoAd's finetuning time is approximately 50 times shorter than OFT's.
> - **Additional Functions**: RoAd not only excels in finetuning efficiency but also features highly efficient batching due to element-wise multiplication. This is not available in OFT, which has batching latency similar to LoRA.
> - **Result**: As demonstrated in Table 2, RoAd significantly outperforms OFT with fewer trainable parameters. OFT with 0.1\% trainable parameters achieves 82.3 on GLUE, while RoAd$_1$ (fc1) with 0.03\% trainable parameters offers 85.1.
>
> $~$
>
> > **W3. Mild overclaim of the results, since there is an outlier on the arithmetic reasoning task for LLaMA2-7B.**
>
> Thank you for your valuable feedback. We apologize for the oversight. In seven out of eight results (as shown in Tables 2, 3, 4, and 13), RoAd demonstrates superior performance with the minimal level of trainable parameters, with the exception of the arithmetic reasoning task on LLaMA2-7B. To improve the accurate statement, we have revised Line 53 to state:
> "Seven out of eight benchmarks indicate that RoAd outperforms other PEFT methods while maintaining a significantly reduced scale of trainable parameters (< 0.1%), as partially depicted in Figure 1."
>
> $~$
>
> > **Q1. Typo**
>
> Thank you for this detailed suggestion. We have corrected this typo in our updated version.
>
>  ---
>
>  $~$
>
> Thank you for your thoughtful suggestions. We have incorporated the new results in the updated version by our side.
>
> In the general rebuttal block, we summarize all new results in our newly uploaded PDF, and highlight some for your easy choice, if you are interested in our response to other reviewers.
>
> If these revisions address your concerns, we kindly request a reconsideration of the scores. Should you have any further questions, we are happy to assist.

---

> > ### Comment · Reviewer_UyLU · 2024-08-14
> >
> > I would like to thank the authors for addressing my concerns. I will keep my original rating of weak accept. I agree that this is a technically solid, moderate-to-high impact paper, with no major concerns with respect to evaluation, resources, reproducibility, ethical considerations.

---

> ### Author Response · Authors · 2024-08-14
> **Thank you for your feedback**
>
> Dear Reviewer UyLU,
>
> Thank you for your feedback. We really enjoy the discussion, and thanks again for your suggestions about:
> 1. Comparing RoAd to FLoRA for the batching efficiency;
> 2. Clear clarification of RoAd's novelty and advantages over OFT;
> 3. Typos and more accurate statement.
>
> We believe these suggestions make our work more solid and strong.
>
> Best!

---

### Official Review · Reviewer_vHYZ · 2024-07-25

**Soundness:** 3
**Presentation:** 3
**Contribution:** 3
**Rating:** 6
**Confidence:** 4

**Summary:**

This paper proposes a novel parameter-efficient fine-tuning method called RoAD, aimed at addressing the challenges of efficient deployment of LLMs that require multiple adapters for distinct needs and enhancing the interpretability of LLMs. The motivation behind this approach stems from the observation that the fine-tuning process primarily affects the angular components of representations rather than their magnitude. Thus, RoAD introduces a 2D rotational approach to representations to achieve parameter-efficient adaptation. Experimental results demonstrate that RoAD achieves a superior accuracy-efficiency trade-off compared to baseline approaches.

**Strengths:**

- Developing effective approaches to adapt the LLM to downstream tasks with better scalability and performance is an important problem. The achieved performance of this paper looks promising to me.

- The paper layout is clear and the writing is easy to understand.

- The observation that during tuning, the directional change is much more significant than the magnitude change is an interesting observation and may further motivate some follow-up research.

**Weaknesses:**

- For the observation section, it would be beneficial to include results from more recent autoregressive language models, such as Llama2, to ensure that the observation is consistent and generalizable across more commonly used models.

- Another concern regarding the observation section is that the metrics used for magnitude and angular change are not identical. As a result, the claim that angular change is larger than magnitude change may be significantly influenced by the chosen metric and scaling. The authors should further justify the validity of this observation when using different metrics.

- The authors should consider further benchmarking the proposed method on generative tasks with instruction tuning and evaluate it on more challenging benchmarks, such as Alpaca-Eval (https://github.com/tatsu-lab/alpaca_eval) and MT-Bench (https://github.com/lm-sys/FastChat/tree/main/fastchat/llm_judge).

**Questions:**

- Can the proposed method be applied to vision language models? Does the observation still stand for vision language models?

**Limitations:**

The authors have mentioned the potential limitations of the paper.

---

> ### Author Rebuttal · Authors · 2024-08-07
>
> Thank you for your time, effort, and thorough review. We appreciate the positive feedback and are encouraged by your highlights:
>
> 1. We develop **effective approaches with better scalability and performance** on an important problem. And the achieved **performance is promising** to you.
> 2. The paper **layout is clear** and the **writing is easy to understand**.
> 3. We offer **interesting observation** that may further **motivate some follow-up research**.
>
> $~$
>
> ---
>
> We address your concerns below:
>
> > **W1. Does the observation from Llama2 stay the same as RoBERTa?**
>
> Thank you for your valuable suggestion. We finetune LLaMA2-7B using LoRA on the GSM8K training set and analyze the magnitude and angle changes of the test set samples, as shown in **Figure A.1\(a)** of the uploaded PDF.
>
> **Summary: LLaMA2-7B exhibits a similar pattern to RoBERTa-base, with more significant changes in angle rather than magnitude.**
>
> - **Experimental Setup:** When directly measuring $\Delta D$ and $\Delta M$, we observe $\Delta D = 1$ and $\Delta M = 0$ for almost all samples, indicating no change in both magnitude and angle. This is because LLaMA2-7B has a much larger hidden size compared to RoBERTa-base (4096 vs. 768) and is more powerful. Finetuning only slightly adapts most dimensions, with significant changes in a limited number of dimensions. Therefore, we apply t-SNE to reduce the dimensions before calculating $\Delta D$ and $\Delta M$.
> - **Detailed Observations:**
>   - Many samples remain close to $\Delta D = 1$ and $\Delta M = 0$, showing no significant change in magnitude or angle.
>   - For samples with changed representations, $\Delta M$ is small (mostly < 0.2), while $\Delta D$ shows a significant change, ranging from 1 to around -0.75.
>   - There are a few outliers with larger changes in magnitude.
>
>
> $~$
>
> > **W2. Is the observation robust across different metrics?**
>
> Thank you for the opportunity to clarify our pilot studies (Section 3.1).
>
> - In our first study, we measure both $\Delta D$ and $\Delta M$ to demonstrate that finetuning primarily affects the angle, as indicated by larger $\Delta D$ values compared to $\Delta M$. Both $\Delta D$ and $\Delta M$ are normalized metrics, with $\Delta D \in [-1, 1]$ and $\Delta M$ representing relative magnitude changes.
> - Acknowledging that our finding might be influenced by the choice of metrics and potentially perceived as subjective, we conduct a second study to disentangle these two factors during finetuning. The results of this second study are consistent with the first, confirming that angle information plays a more crucial role than magnitude in finetuning.
>
> These studies collectively ensure that our observations are robust and supported by practical experiments. If there are specific metrics you would like us to consider, please let us know, and we would be happy to provide additional analyses using those metrics.
>
> $~$
>
> > **W3. Further benchmark RoAd, like with AlpacaEval, MT-Bench.**
>
> Thank you for this great suggestion. We benchmark RoAd using AlpacaEval2.0, and the results can be found in **Table A.1** of the attached PDF. Due to time and resource constraints, benchmarking with MT-Bench is on our to-do list.
>
> **Summary: RoAd demonstrates superior performance compared to all baselines, while utilizing the least number of trainable parameters.**
>
> - **Experimental Setup:** We finetune LLaMA2-7B with two instruction-tuning datasets and evaluate the model using AlpacaEval2.0. This evaluation employs GPT-4 to assess the responses generated by the finetuned model against those produced by Text-davinci-003. We don't choose GPT-4 as the reference model, because GPT-4 is too powerful than LLaMA2-7B. The proof-of-concept experiment with LoRA shows the win-rate < 5\%.
>
>
> $~$
>
> > **Q1. Apply RoAd to vision language models, and its observation.**
>
> Thank you for your suggestion. We finetune LLaVA-1.5-7B using RoAd, and the results are presented in **Table A.2** of the uploaded PDF.
>
> **Summary: RoAd achieves the same average performance as LoRA with only 1/4 of its trainable parameters.**
>
> - **Experimental Setup:** [R1] requires 4.61% trainable parameters for LoRA on this task, while most tasks with LoRA in our paper need < 1%, showing that this task is knowledge-intensive. Therefore, we need to scale RoAd's trainable parameters. For this purpose, we combine it with LoRA due to the limited number of $\theta_i$ and $\alpha_i$ in $R$. The combination is represented as $Z = (RW + BA)^TX$, where $R$ is the rotation matrix from RoAd, and $A$ and $B$ are from LoRA. We adjust the LoRA rank to vary the number of trainable parameters. We combine RoAd$_1$ with LoRA, but not RoAd$_2$ or RoAd$_4$, as their primary design purpose is to increase the number of trainable parameters.
> - **Results:**
>     - With only 0.08% trainable parameters, RoAd$_4$ already achieves 96.9% (66.4/68.5) of the accuracy of LoRA with 4.61% trainable parameters. By combining RoAd$_1$ with LoRA, we achieve the same performance as LoRA with just 1/4 of its trainable parameters. This demonstrates RoAd's excellent scalability when combined with LoRA. Further promising scaling results can be found in our response to Reviewer zyNQ in Weakness 2, if you are interested.
>     - The observations are very similar to our response to Weakness 1. We also need to apply t-SNE to reduce the dimensions.
>
>
> $~$
>
> [R1] Visual Instruction Tuning, Haotian Liu, etc.
>
> ---
>
> $~$
>
> Thank you for your thoughtful suggestions. We have incorporated the new results in the updated version by our side.
>
> In the general rebuttal block, we summarize all new results in our newly uploaded PDF, and highlight some for your easy choice, if you are interested in our response to other reviewers.
>
> If these revisions address your concerns, we kindly request a reconsideration of the scores. Should you have any further questions, we are happy to assist.

---

> > ### Comment · Reviewer_vHYZ · 2024-08-12
> >
> > I would like to thank the authors for their detailed responses and have raised my score accordingly.

---

> ### Author Response · Authors · 2024-08-12
> **Thank you for your raised score**
>
> Dear Reviewer vHYZ,
>
> We are very encouraged by your increased score.
>
> We really enjoy the discussion, and thank you for your suggestions about:
> 1. Observation from Llama2;
> 2. Further benchmarking RoAd with AlpacaEval and MT-Bench, and on visual instruction tasks.
>
> We believe these suggestions make our work more solid and strong.
>
> Best!

---

### Author Rebuttal · Authors · 2024-08-07

Here we summarize the new results in the uploaded PDF, you can selectively read them if you are interested. We highlight some for your easy choice.

| Table or Figure | Content | Where for details (i.e. response to which point of which reviewer) |
| :---  |  :--- | :--- |
| Table A.1 | **Further benchmark RoAd with AlpacaEval2.0** | W3 of Reviewer vHYZ |
| Table A.2 | **Further benchmark RoAd with visual instruction tuning** | Q1 of Reviewer vHYZ |
| Table A.3 (Row 1-10) | RoAd$_2$ and RoAd$_4$ on GLUE | W3 of Reviewer zyNQ |
| Table A.3 (Row 11-13) | RoAd's multitasking ability | W1 of Reviewer iQCQ |
| Table A.4 and Table A.5 (Row 1-2) | **RoAd's scalability** | W2 of Reviewer zyNQ |
| Table A.4 and Table A.5 (Row 3-4) | New baseline, SSF | W6 of Reviewer iQCQ |
| Figure A.1 (a) | Pilot study on LLaMA2-7B | W1 of Reviewer vHYZ |
|  Figure A.1 (b) | More batching efficiency results | W1 of Reviewer UyLU and W5 of Reviewer iQCQ |
| Figure A.1 (c) | **An overview diagram** | W4 of Reviewer iQCQ |

---

### Author Response · Authors · 2024-08-10
**Kind reminder for your feedback**

Dear Reviewers,

Thanks again for the reviews! We deeply appreciate the time and effort you have invested in evaluating our work. We really value your reviews, and have carefully considered your comments and prepared a detailed rebuttal addressing each point raised.

We are hoping you would be able to review our responses and to engage in a discussion before the end of the discussion period. Please let us know if there are any specific areas you'd like us to clarify further or discuss in more detail.

Thank you once again for your commitment to the review process.

Best regards,

Authors

---

### Decision · Program_Chairs · 2024-09-25

**Decision:**

Accept (poster)

**Comment:**

This paper introduces a novel parameter-efficient finetuning method named RoAd, which employs a 2D rotation to adapt large language models (LLMs). The paper addresses three key challenges in parameter-efficient finetuning (PEFT): efficient deployment of LLMs that require multiple task-specific adapters, batching efficiency, and interpretability. RoAd demonstrates its effectiveness across various benchmarks, including GLUE, commonsense reasoning, and arithmetic reasoning tasks, with less than 0.1% trainable parameters. The authors provided a thorough experimental analysis, showing RoAd’s superior performance over baseline methods like LoRA and OFT.
The initial reviews were mixed, with the reviewers recognizing the significance of the problem and the promising results but raising concerns about the evaluation, novelty over existing methods, and clarity of the paper’s structure. The authors engaged actively with the reviewers, providing detailed rebuttals and additional experiments that addressed most of the concerns. After the rebuttal, the final ratings are uniformly positive, with all reviewers agreeing that the paper is technically solid and has moderate-to-high impact. The reviewers acknowledged the authors' efforts to address their concerns, which led to increased scores in some cases. The AC recommends accepting this paper, with the expectation that the authors will incorporate the final revisions to further strengthen the paper's clarity and completeness.